# Unbalancedness in Neural Monge Maps Improves Unpaired Domain Translation

**Luca Eyring**[1,2,3*]    **Dominik Klein**[2,3*]    **Théo Uscidda**[2,4*]    **Giovanni Palla**[2,3]
**Niki Kilbertus**[2,3,5]    **Zeynep Akata**[1,2,3]    **Fabian Theis**[2,3,5]

[1]Tübingen AI Center    [2]Helmholtz Munich    [3]TU Munich
[4]CREST-ENSAE    [5]Munich Center for Machine Learning (MCML)
`luca.eyring@uni-tuebingen.de`    `theo.uscidda@ensae.fr`
`{dominik.klein,fabian.theis}@helmholtz-muenchen.de`

## Abstract

In optimal transport (OT), a Monge map is known as a mapping that transports a source distribution to a target distribution in the most cost-efficient way. Recently, multiple neural estimators for Monge maps have been developed and applied in diverse unpaired domain translation tasks, e.g. in single-cell biology and computer vision. However, the classic OT framework enforces mass conservation, which makes it prone to outliers and limits its applicability in real-world scenarios. The latter can be particularly harmful in OT domain translation tasks, where the relative position of a sample within a distribution is explicitly taken into account. While unbalanced OT tackles this challenge in the discrete setting, its integration into neural Monge map estimators has received limited attention. We propose a theoretically grounded method to incorporate unbalancedness into *any* Monge map estimator. We improve existing estimators to model cell trajectories over time and to predict cellular responses to perturbations. Moreover, our approach seamlessly integrates with the OT flow matching (OT-FM) framework. While we show that OT-FM performs competitively in image translation, we further improve performance by incorporating unbalancedness (UOT-FM), which better preserves relevant features. We hence establish UOT-FM as a principled method for unpaired image translation.

## 1 Introduction

Unpaired domain translation aims to transform data from a source to a target distribution without access to paired training samples. This setting poses the significant challenge of achieving a meaningful translation between distributions while retaining relevant input features. Although there are many ways to define the desired properties of such a transformation, optimal transport (OT) offers a natural framework by matching samples across distributions in the most cost-efficient way. If this optimal correspondence can be formulated as a map, such a map is known as a Monge map.

Recently, a considerable number of neural parameterizations to estimate Monge maps have been proposed. While earlier estimators were limited to the squared Euclidean distance (Makkuva et al., 2020; Korotin et al., 2020; Amos, 2022), more flexible approaches have been proposed recently (Uscidda & Cuturi, 2023; Tong et al., 2023b; Pooladian et al., 2023a;b). Neural Monge maps have been successfully applied to a variety of domain translation tasks including applications in computer vision (Korotin et al., 2022; Tong et al., 2023b; Pooladian et al., 2023a; Mokrov et al., 2023) and the modeling of cellular responses to perturbations (Bunne et al., 2021).

In its original formulation, optimal transport assumes static marginal distributions. This can limit its applications as it cannot account for **[i]** outliers and **[ii]** undesired distribution shifts, e.g. class imbalance between distributions as visualized in Figure 1. Unbalanced OT (UOT) (Chizat et al., 2018a) overcomes these limitations by replacing the conservation of mass constraint with a penalization on mass deviations. The practical significance of unbalancedness in discrete OT has been demonstrated

---

*equal contribution

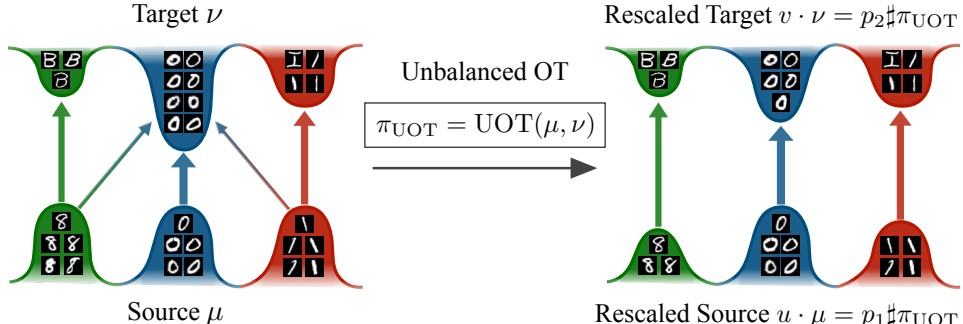

Figure 1: Comparison of balanced and unbalanced Monge map computed on the EMNIST dataset translating *digits → letters*. Source and target distribution are rescaled leveraging the unbalanced OT coupling. The computed balanced mapping includes $8 \rightarrow \{O, B\}$, and $1 \rightarrow \{O, I\}$ because of the distribution shift between digits and letters. With unbalancedness $8 \rightarrow B$, and $1 \rightarrow I$ are recovered.

in various applications, e.g. in video registration (Lee et al., 2020), computer vision (Plaen et al., 2023), or domain adaptation (Fatras et al., 2021a). Existing methods for estimating neural Monge maps with unbalancedness (Yang & Uhler, 2019; Lübeck et al., 2022) are limited to specific Monge map estimators and rely on adversarial training, see Section 4 for a detailed discussion.

In light of these limitations, we introduce a new framework for incorporating unbalancedness into *any* Monge map estimator based on a re-scaling scheme. We motivate our approach theoretically by proving that we can incorporate unbalancedness into neural Monge maps by rescaling source and target measures accordingly. To validate the versatility of our approach, we showcase its applicability on both synthetic and real-world data, utilizing different neural Monge map estimators. We highlight the critical role of incorporating unbalancedness to infer trajectories in developmental single-cell data using ICNN-based estimators (OT-ICNN) (Makkuva et al., 2020) and to predict cellular responses to perturbations with the Monge gap (Uscidda & Cuturi, 2023).

Monge maps can also be approximated with OT flow matching (OT-FM) (Lipman et al., 2023; Liu et al., 2022; Albergo & Vanden-Eijnden, 2023), a simulation-free technique for training continuous normalizing flows (Chen et al., 2018) relying on mini-batch OT couplings (Pooladian et al., 2023a; Tong et al., 2023b). The universal applicability of our method allows us to extend OT-FM to the unbalanced setting (UOT-FM). We demonstrate that unbalancedness is crucial for obtaining meaningful matches when translating *digits* to *letters* in the EMNIST dataset (Cohen et al., 2017). Additionally, we benchmark OT-FM on unpaired natural image translation and show that it achieves competitive results compared to established methods. Moreover, UOT-FM elucidates the advantages of unbalancedness in image translation as it **[i]** improves overall performance upon OT-FM while additionally **[ii]** helping to preserve relevant input features and lowering the learned transport cost. This establishes UOT-FM as a new principled method for unpaired image translation. To summarize:

1. We propose an efficient algorithm to integrate *any* balanced Monge map estimator into the unbalanced OT framework.
2. We theoretically verify our approach by proving that computing a Monge map between measures of unequal mass can be reformulated as computing a Monge map between two rescaled measures.
3. We demonstrate that incorporating unbalancedness yields enhanced results across three distinct tasks employing three different Monge map estimators. We find that our proposed approach enables the recovery of more biologically plausible cell trajectories and improves the prediction of cellular responses to cancer drugs. Furthermore, our method helps to preserve relevant input features on unpaired natural image translation.

## 2 BACKGROUND

**Notation.** For $\Omega \subset \mathbb{R}^d$, $\mathcal{M}^+(\Omega)$ and $\mathcal{M}_1^+(\Omega)$ are respectively the set of positive measures and probability measures on $\Omega$. For $\mu \in \mathcal{M}^+(\Omega)$, we write $\mu \ll \mathcal{L}_d$ if $\mu$ is absolutely continuous w.r.t. the Lebesgue measure. For a Lebesgue measurable map $T : \Omega \rightarrow \Omega$ and $\mu \in \mathcal{M}^+(\Omega)$, $T\sharp\mu$ denotes the pushforward of $\mu$ by $T$, namely, for all Borel sets $A \subset \Omega$, $T\sharp\mu(A) = \mu(T^{-1}(A))$. For $\mu, \nu \in \mathcal{M}^+(\Omega)$, $\Pi(\mu, \nu) := \{\pi : p_1\sharp\pi = \mu, p_2\sharp\pi = \nu\} \subset \mathcal{P}(\Omega \times \Omega)$ where $p_1 : (\mathbf{x}, \mathbf{y}) \mapsto \mathbf{x}$ and $p_2 : (\mathbf{x}, \mathbf{y}) \mapsto \mathbf{y}$ are the canonical projectors, so $p_1\sharp\pi$ and $p_2\sharp\pi$ are the marginals of $\pi$.

**Monge and Kantorovich Formulations.** Let $\Omega \subset \mathbb{R}^d$ be a compact set and $c : \Omega \times \Omega \to \mathbb{R}$ a continuous cost function. The Monge problem (MP) between $\mu, \nu \in \mathcal{M}_1^+(\Omega)$ consists of finding a map $T : \Omega \to \Omega$ that push-forwards $\mu$ onto $\nu$, while minimizing the average displacement cost

$$\inf_{T:T\sharp\mu=\nu} \int_\Omega c(\mathbf{x}, T(\mathbf{x})) \, \mathrm{d}\mu(\mathbf{x}) \,. \tag{MP}$$

We call any solution $T^\star$ to Problem (MP) a Monge map between $\mu$ and $\nu$. Solving this problem is difficult, especially for discrete $\mu, \nu$, for which the constraint set can even be empty. Instead of transport maps, the Kantorovich problem (KP) of OT considers couplings $\pi \in \Pi(\mu, \nu)$:

$$\mathrm{W}_c(\mu, \nu) := \min_{\pi \in \Pi(\mu, \nu)} \int_{\Omega \times \Omega} c(\mathbf{x}, \mathbf{y}) \, \mathrm{d}\pi(\mathbf{x}, \mathbf{y}) \,. \tag{KP}$$

An OT coupling $\pi_{\mathrm{OT}}$, a solution of (KP), always exists. When Problem (MP) admits a solution, these formulations coincide with $\pi_{\mathrm{OT}} = (\mathrm{I}_d, T^\star)\sharp\mu$, i.e. $\pi_{\mathrm{OT}}$ is deterministic, which means that if $(\mathbf{x}, \mathbf{y}) \sim \pi$, then $\mathbf{y} = T(\mathbf{x})$. Problem (KP) admits a dual formulation. Denoting by $\Phi_c(\mu, \nu) = \{(f, g) | f, g : \Omega \to \mathbb{R}, f \oplus g \leq c, \mu \otimes \nu\text{-a.e.}\}$ where $f \oplus g : (\mathbf{x}, \mathbf{y}) \mapsto f(\mathbf{x}) + g(\mathbf{y})$ is the tensor sum, it reads

$$(f, g) \in \operatorname*{arg\,min}_{(f,g) \in \Phi_c(\mu,\nu)} \int_\Omega f(\mathbf{x}) \, \mathrm{d}\mu(\mathbf{x}) + \int_\Omega g(\mathbf{y}) \, \mathrm{d}\nu(\mathbf{y}) \,. \tag{DKP}$$

**Extension to the Unbalanced Setting.** The Kantorovich formulation imposes mass conservation, so it cannot handle arbitrary positive measures $\mu, \nu \in \mathcal{M}^+(\Omega)$. Unbalanced optimal transport (UOT) (Benamou, 2003; Chizat et al., 2018a) lifts this constraint by penalizing instead the deviation of $p_1\sharp\pi$ to $\mu$ and $p_2\sharp\pi$ to $\nu$ via a $\phi$-divergence $\mathrm{D}_\phi$. These divergences are defined trhough an entropy function $\phi : (0, +\infty) \to [0, +\infty]$, which is convex, positive, lower-semi-continuous and s.t. $F(1) = 0$. Denoting $\phi'_\infty = \lim_{x\to\infty} \phi(\mathbf{x})/\mathbf{x}$ its recession constant, $\alpha, \beta \in \mathcal{M}(\Omega)$, we have

$$\mathrm{D}_\phi(\alpha|\beta) = \int_\Omega \phi\left(\frac{\mathrm{d}\alpha}{\mathrm{d}\beta}\right) \mathrm{d}\alpha + \phi'_\infty \int_\Omega \mathrm{d}\alpha^\perp \,, \tag{1}$$

where we write the Lebesgue decomposition $\alpha = \frac{\mathrm{d}\alpha}{\mathrm{d}\beta}\beta + \alpha^\perp$, with $\frac{\mathrm{d}\alpha}{\mathrm{d}\beta}$ the relative density of $\alpha$ w.r.t. $\beta$. In this work, we consider strictly convex and differentiable entropy functions $\phi$, with $\phi'_\infty = +\infty$. This includes, for instance, the KL, the Jensen-Shanon, or the $\chi^2$ divergence. Introducing $\lambda_1, \lambda_2 > 0$ controlling how much mass variations are penalized as opposed to transportation, the unbalanced Kantorovich problem (UKP) then seeks a measure $\pi \in \mathcal{M}^+(\mathcal{X} \times \mathcal{Y})$:

$$\mathrm{UW}_c(\mu, \nu) := \min_{\pi \in \mathcal{M}^+(\Omega \times \Omega)} \int_{\Omega \times \Omega} c(\mathbf{x}, \mathbf{y}) \, \mathrm{d}\pi(\mathbf{x}, \mathbf{y}) + \lambda_1 \mathrm{D}_\phi(p_1\sharp\pi|\mu) + \lambda_2 \mathrm{D}_\phi(p_2\sharp\pi|\nu) \,. \tag{UKP}$$

An UOT coupling $\pi_{\mathrm{UOT}}$ always exists. In practice, instead of directly selecting $\lambda_i$, we introduce $\tau_i = \frac{\lambda_i}{\lambda_i + \varepsilon}$, where $\varepsilon$ is the entropy regularization parameter as described in Appendix B.1. Then, we recover balanced OT for $\tau_i = 1$, equivalent to $\lambda_i \to +\infty$, and increase unbalancedness by decreasing $\tau_i$. We write $\tau = (\tau_1, \tau_2)$ accordingly. Finally, the (UKP) also admits a dual formulation which reads

$$(f, g) \in \operatorname*{arg\,min}_{(f,g) \in \Phi_c(\mu,\nu)} \int_\Omega -\phi^*(-f(\mathbf{x})) \, \mathrm{d}\mu(\mathbf{x}) + \int_\Omega -\phi^*(g(\mathbf{y})) \, \mathrm{d}\nu(\mathbf{y}) \,. \tag{UDKP}$$

## 3 Unbalanced Neural Monge Maps

**Unbalanced Monge Maps.** Although the Kantorovich formulation has an unbalanced extension, deriving an analogous Monge formulation that relaxes the mass conservation constraint is more challenging. Indeed, by adopting the same strategy of replacing the marginal constraint $T\sharp\mu = \nu$ with a penalty $\mathrm{D}_\phi(T\sharp\mu, \nu)$, we would allow the distribution of transported mass $T\sharp\mu$ to shift away from $\nu$, but the total amount of transported mass would still remain the same. Instead, we can follow a different approach and consider a two-step procedure where (i) we re-weight $\mu$ and $\nu$ into measures $\tilde\mu = u \cdot \mu$ and $\tilde\nu = v \cdot \mu$ having the same total mass using $u, v : \mathbb{R}^d \to \mathbb{R}^+$, and (ii) then compute a (balanced) Monge map between $\tilde\mu$ and $\tilde\nu$. Hence, it remains to define the re-weighted measures $\tilde\mu, \tilde\nu$. Therefore, we seek to achieve two goals: **[i]** create or destroy mass to minimize the cost of transporting measure $\tilde\mu$ to $\tilde\nu$, while **[ii]** being able to control the deviation of the latter from the input measures $\mu$ and $\nu$. To that end, we can show that computing $\pi_{\mathrm{UOT}}$ between $\mu$ and $\nu$ implicitly amounts to following this re-balancing procedure. Additionally, $\tilde\mu$ and $\tilde\nu$ can be characterized precisely.

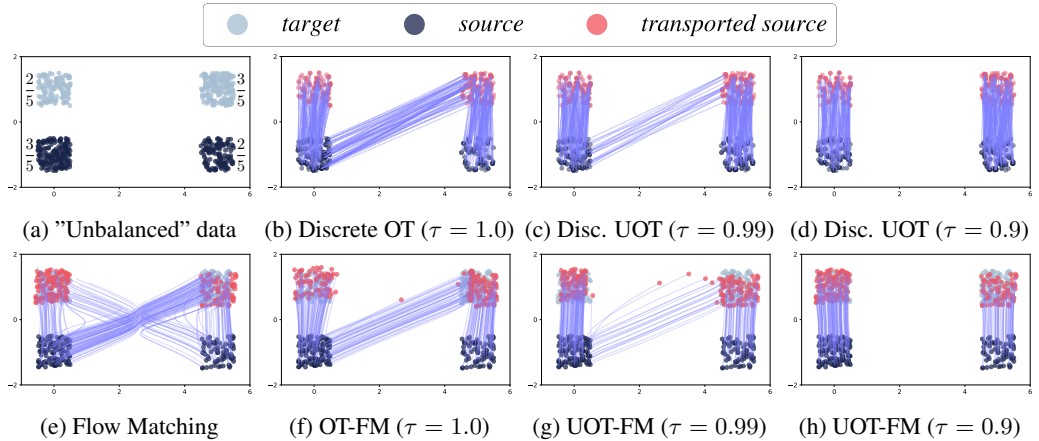

(a) "Unbalanced" data    (b) Discrete OT ($\tau = 1.0$)    (c) Disc. UOT ($\tau = 0.99$)    (d) Disc. UOT ($\tau = 0.9$)

(e) Flow Matching    (f) OT-FM ($\tau = 1.0$)    (g) UOT-FM ($\tau = 0.99$)    (h) UOT-FM ($\tau = 0.9$)

Figure 2: Different maps on data drawn from a mixture of uniform distribution, where the density in the bottom left and the top right ($\frac{3}{5}$) is higher than in the top left and bottom right ($\frac{2}{5}$) (Appendix F.6). Besides the data in Figure 2a, the first row shows results of discrete balanced OT (2b), and discrete unbalanced OT with two different degrees of unbalancedness $\tau$ (2c, 2d). The second row shows the maps obtained by FM with independent coupling (2e), OT-FM (2f), and UOT-FM (2g, 2h).

**Proposition 3.1** (Re-balancing the UOT problem). *Let $\pi_{\mathrm{UOT}}$ be the solution of problem (UKP) between $\mu, \nu \in \mathcal{M}^+(\Omega)$, for $\tau_1, \tau_2 > 0$. Then, the following holds:*

1. *$\pi_{\mathrm{UOT}}$ solves the balanced problem (KP) between its marginal $\tilde{\mu} = p_1 \sharp \pi_{\mathrm{UOT}}$ and $\tilde{\nu} = p_2 \sharp \pi_{\mathrm{UOT}}$, which are re-weighted versions of $\mu$ an $\nu$ that have the same total mass. Indeed, $\tilde{\mu} = \bar{\phi}(-f^\star) \cdot \mu$ and $\tilde{\nu} = \bar{\phi}(-g^\star) \cdot \nu$, where $\bar{\phi} = (\phi^*)'$ and $f^\star, g^\star : \Omega \to \mathbb{R}$ are solution of (UDKP).*

2. *If we additionally assume that $\mu \ll \mathcal{L}_d$ and that the cost $c(\mathbf{x}, \mathbf{y}) = h(\mathbf{x} - \mathbf{y})$ with $h$ strictly convex, then $\pi_{\mathrm{UOT}}$ is unique and deterministic: $\pi_{\mathrm{UOT}} = (\mathrm{I}_d, T^\star) \sharp \tilde{\mu}$, where $T^\star = \mathrm{I}_d - \nabla h^* \circ f^\star$ is the Monge map between $\tilde{\mu}$ and $\tilde{\nu}$ for cost $c$.*

**Remark 3.2.** *All the cost functions $c(\mathbf{x}, \mathbf{y}) = \|\mathbf{x} - \mathbf{y}\|_2^p$ with $p > 1$ can be dealt with via point 2 of Prop. 3.1. For $p = 2$, we recover an unbalanced counterpart of Brenier (1987) Theorem stating that $\pi_{\mathrm{UOT}}$ is supported on the graph of $T = \nabla \varphi^\star$ where $\varphi^\star : \mathbb{R}^d \to \mathbb{R}$ is convex.*

Prop. 3.1 highlights that the optimal re-weighting procedure relies on setting $\tilde{\mu} = p_1 \sharp \pi_{\mathrm{UOT}}$ and $\tilde{\nu} = p_2 \sharp \pi_{\mathrm{UOT}}$, where we directly control the deviation to the input measures $\mu$ and $\nu$ with $\tau_1$ and $\tau_2$. This enables us to define *unbalanced Monge maps* formally. We illustrate this concept in Figure 1.

**Definition 3.3** (Unbalanced Monge maps). *Provided that it exists, we define the unbalanced Monge map between $\mu, \nu \in \mathcal{M}^+(\Omega)$ as the Monge map $T^\star$ between the marginal $\tilde{\mu}$ and $\tilde{\nu}$ of the $\pi_{\mathrm{UOT}}$ between $\mu$ and $\nu$. From point 2 of Prop. 3.1, it satisfies $\pi_{\mathrm{UOT}} = (\mathrm{I}_d, T^\star) \sharp \tilde{\mu}$.*

Proofs are provided in Appendix A. We stress that even when $\mu$ and $\nu$ have the same mass, the unbalanced Monge map does not coincide with the classical Monge map as defined in (MP). Indeed, $\tilde{\mu} \neq \mu$ and $\tilde{\nu} \neq \nu$ unless we impose $\tau_1 = \tau_2 = 1$ to recover each marginal constraint.

### 3.1 Estimation of Unbalanced Monge Maps

Learning an unbalanced Monge map between $\mu, \nu$ remains to learn a balanced Monge map between $\tilde{\mu} = u \cdot \mu$, $\tilde{\nu} = v \cdot \nu$. Provided that we can sample from $\tilde{\mu}$ and $\tilde{\nu}$, this can be done with any Monge map estimator. In practice, we can produce approximate samples from the re-weighted measures using samples from the input measure $\mathbf{x}_1 \ldots \mathbf{x}_n \sim \mu$ and $\mathbf{y}_1 \ldots \mathbf{y}_n \sim \nu$ (Fatras et al., 2021b). First, **[i]** we compute $\hat{\pi}_{\mathrm{UOT}} \in \mathbb{R}_+^{n \times n}$ the solution of problem (UKP) between $\hat{\mu}_n = \frac{1}{n} \sum_{i=1}^n \delta_{\mathbf{x}_i}$ and $\hat{\nu}_n = \frac{1}{n} \sum_{i=1}^n \delta_{\mathbf{y}_i}$, then **[ii]** we sample $(\tilde{\mathbf{x}}_1, \tilde{\mathbf{y}}_1), \ldots, (\tilde{\mathbf{x}}_n, \tilde{\mathbf{y}}_n) \sim \hat{\pi}_{\mathrm{UOT}}$. Indeed, if we set $\mathbf{a} := \hat{\pi}_{\mathrm{UOT}} \mathbf{1}_n$ and $\mathbf{b} := \hat{\pi}_{\mathrm{UOT}}^\top \mathbf{1}_n$, and define $\tilde{\mu}_n := \sum_{i=1}^n a_i \delta_{\mathbf{x}_i} = p_1 \sharp \pi_{\mathrm{UOT}}$ and $\tilde{\nu}_n := \sum_{i=1}^n b_i \delta_{\mathbf{y}_i} = p_2 \sharp \pi_{\mathrm{UOT}}$, these are empirical approximation of $\tilde{\mu}$ and $\tilde{\nu}$, and $\tilde{\mathbf{x}}_1 \ldots \tilde{\mathbf{x}}_n \sim \tilde{\mu}$ and $\tilde{\mathbf{y}}_1 \ldots \tilde{\mathbf{y}}_n \sim \tilde{\nu}$. Such couplings $\hat{\pi}_{\mathrm{UOT}}$ can be estimated efficiently using entropic regularization (Cuturi, 2013; Séjourné et al., 2022) (Appendix B.1). Moreover, we can learn the re-weighting functions as we have access to estimates of their pointwise evaluation: $u(\mathbf{x}_i) \approx n \cdot a_i = (\mathrm{d}\tilde{\mu}_n / \mathrm{d}\hat{\mu}_n)(\mathbf{x}_i)$ and $v(\mathbf{y}_i) \approx n \cdot b_i =$

$(\mathrm{d}\tilde{\nu}_n / \mathrm{d}\hat{\nu}_n)(\mathbf{y}_i)$. We provide our general training procedure in Appendix D and note that UOT introduces the hyperparameter $\tau$, which controls the level of unbalancedness as visualized in Figure 2. The main limitation of our proposed framework revolves around the choice of $\tau$, which as discussed in Séjourné et al. (2023) is not always evident and thereby facilitates a grid search (Appendix B.2).

**Merits of Unbalancedness.** In addition to removing outliers and addressing class imbalances across the entire dataset, as mentioned in Section 1, the batch-wise training approach employed by many neural Monge map estimators makes unbalancedness a particularly favorable characteristic. Specifically, the distribution within a batch is likely not reflective of the complete source and target distributions, leading to a class imbalance at the batch level. Similarly, a data point considered normal in the overall distribution may be treated as an outlier within a batch. Consequently, the use of unbalanced Monge maps serves to mitigate the risk of suboptimal pairings within individual batches, which can help to stabilize training and speed up convergence (Fatras et al., 2021a; Choi et al., 2023). Next, we introduce the three estimators that we employ to compute unbalanced Monge maps.

**OT-ICNN.** When $c(\mathbf{x}, \mathbf{y}) = \|\mathbf{x} - \mathbf{y}\|_2^2$, Brenier (1987)'s Theorem states that $T^\star = \nabla \varphi^\star$, where $\varphi^\star : \Omega \to \mathbb{R}$ is convex and solves a reformulation of the dual problem (DKP)

$$\varphi^\star \in \arg\inf_{\varphi \text{ convex}} \int_\Omega \varphi(\mathbf{x}) \, \mathrm{d}\mu(\mathbf{x}) + \int_\Omega \varphi^*(\mathbf{y}) \, \mathrm{d}\mu(\mathbf{y}), \qquad (2)$$

where $\varphi^*$ denotes the convex conjugate of $\varphi$. Makkuva et al. (2020) propose to solve Eq. 2 using Input Convex Neural Networks (ICNNs) (Amos et al., 2017), which are parameterized convex functions $\mathbf{x} \mapsto \varphi_\theta(\mathbf{x})$. To avoid the explicit computation of the convex conjugate $\varphi_\theta^*$, they approximate it by parameterizing an additional ICNN $\eta_\theta$ and solve

$$\sup_{\eta_\theta \text{ ICNN}} \inf_{\varphi_\theta \text{ ICNN}} \mathcal{L}_{\text{OT-ICNN}}(\theta) := \int_{\Omega \times \Omega} (\eta_\theta(\nabla\varphi_\theta(\mathbf{x})) - \langle \mathbf{x}, \nabla\varphi_\theta(\mathbf{x}) \rangle - \eta_\theta(\mathbf{y})) \, \mathrm{d}\mu(\mathbf{x}) \, \mathrm{d}\nu(\mathbf{y}), \quad (3)$$

where an estimate of the Monge map is recovered through $T_\theta = \nabla\varphi_\theta$. We refer to this estimation procedure as OT-ICNN and use it to learn unbalanced Monge maps (UOT-ICNN) in Section 5.1.

**Monge Gap.** ICNN-based methods leverage Brenier (1987)'s Theorem, that can only be applied when $c(\mathbf{x}, \mathbf{y}) = \|\mathbf{x} - \mathbf{y}\|_2^2$. Recently, Uscidda & Cuturi (2023) proposed a method to learn Monge maps for any differentiable cost. They define a regularizer $\mathcal{M}_\mu^c$, called the Monge gap, that quantifies the lack of Monge optimality of a map $T : \mathbb{R}^d \to \mathbb{R}^d$. More precisely,

$$\mathcal{M}_\mu^c(T) = \int_\Omega c(\mathbf{x}, \mathbf{y}) \, \mathrm{d}\mu(\mathbf{x}) - W_c(\mu, T\sharp\mu). \qquad (4)$$

From the definition, $\mathcal{M}_\mu^c(T) \geq 0$ with equality i.f.f. $T$ is a Monge map between $\mu$ and $T\sharp\mu$ for the cost $c$. Then, the Monge gap can be used with any divergence $\Delta$ to define the loss

$$\mathcal{L}_{\text{OT-MG}}(\theta) := \Delta(T_\theta\sharp\mu, \nu) + \mathcal{M}_\mu^c(T_\theta) \qquad (5)$$

which is 0 i.f.f. $T_\theta$ is a Monge map between $\mu$ and $\nu$ for cost $c$. We refer to the estimation procedure minimizing this loss as OT-MG and use it to learn unbalanced Monge maps (UOT-MG) in Section 5.2.

**Flow Matching.** Continuous Normalizing Flows (CNFs) (Chen et al., 2018) are a family of continuous-time deep generative models that construct a probability path between $\mu$ and $\nu$ using the flow $(\phi_t)_{t \in [0,1]}$ induced by a neural velocity field $(v_{t,\theta})_{t \in [0,1]}$. Flow Matching (FM) (Lipman et al., 2023; Liu et al., 2022; Albergo & Vanden-Eijnden, 2023) is a simulation-free technique to train CNFs by constructing probability paths between individual data samples $\mathbf{x}_0 \sim \mu$, $\mathbf{x}_1 \sim \nu$, and minimizing

$$\mathcal{L}_{\text{FM}}(\theta) = \mathbb{E}_{t,(\mathbf{x}_0,\mathbf{x}_1)\sim\pi_{\text{ind}}} \|v_\theta(t, \mathbf{x}_t) - (\mathbf{x}_1 - \mathbf{x}_0)\|_2^2, \qquad (6)$$

where $\pi_{\text{ind}} = \mu \otimes \nu$. When this loss is zero, Lipman et al. (2023, Theorem 1) states that $\phi_1$ is push-forward map between $\mu$ and $\nu$, namely $\phi_1\sharp\mu = \nu$. While here FM yields individual straight paths between source and target pairs, the map $\phi_1$ is not a Monge map. OT-FM (Pooladian et al., 2023a; Tong et al., 2023b) suggests to use an OT coupling $\pi_{\text{OT}}$ instead of $\pi_{\text{ind}}$ in the FM loss, which in practice is approximated batch-wise. It can then be shown that $\psi_1$ approximates a Monge map asymptotically (Pooladian et al., 2023a, Theorem 4.2). Thanks to Proposition 3.1, we could extend OT-FM to the unbalanced Monge map setting (UOT-FM) by following Algorithm 1 and rescaling the measures batch-wise before applying OT-FM. In this special case however, both the unbalanced rescaling *and* the coupling computation can be done in one step by leveraging the unbalanced coupling $\pi_{\text{UOT}}$, which again is approximated batch-wise. The full algorithm is described in Appendix D.

## 4 RELATED WORK

While the majority of literature in the realm of neural OT considers the balanced setting (Makkuva et al., 2020; Korotin et al., 2021; Uscidda & Cuturi, 2023; Tong et al., 2023b; Korotin et al., 2022), only a few approaches have been proposed to loosen the strict marginal constraints. Gazdieva et al. (2023) consider partial OT, a less general problem setting than unbalanced OT. Yang & Uhler (2019) propose a GAN-based approach for unbalanced OT plans (as opposed to deterministic maps) and very recently, Choi et al. (2023) employ a methodology based on the semi-dual formulation for generative modeling of images. Finally, Lübeck et al. (2022) introduced a way to incorporate unbalancedness in ICNN-OT. In the following, we highlight the main differences to our approach.

First, our proposed method can be applied to *any* neural Monge Map estimator, while the method proposed in Lübeck et al. (2022) requires the explicit parameterization of the inverse Monge map. While this requirement is satisfied in ICNN-OT, more flexible and performant neural Monge map estimators like MG-OT (Uscidda & Cuturi, 2023) and OT-FM (Pooladian et al., 2023a; Tong et al., 2023b) do not model the inverse Monge map. Second, Lübeck et al. (2022) suggest rescaling the distributions based on the solution of a discrete unbalanced OT coupling between the push-forward of the source distribution and the target distribution, while we consider unbalancedness between the source and the target distribution. This entails that their rescaled marginals are not optimal (Proposition 3.1). Third, our approach is computationally more efficient, as our method requires the computation of at most one discrete unbalanced OT plan, while the algorithm proposed in Lübeck et al. (2022) includes the computation of two discrete unbalanced OT plans. Related work for each of the specific domain translation tasks is discussed in Appendix E due to space restrictions.

## 5 EXPERIMENTS

We demonstrate the importance of learning unbalanced Monge maps on three different domain translation tasks leveraging three different (balanced) Monge map estimators, which showcases the flexibility of our proposed method. To start with, we demonstrate the necessity of incorporating unbalancedness when performing trajectory inference on single-cell RNAseq data, which we use ICNN-OT for. Subsequently, we improve the predictive performance of modeling cellular responses to perturbations with an unbalanced Monge map building upon MG-OT. Both of these tasks can be formulated as an unpaired single-cell to single-cell translation task. Finally, we show in Section 5.3 that OT-FM performs competitively on unpaired image-to-image translation tasks, while its unbalanced formulation (UOT-FM) further improves performance and helps to preserve characteristic input features. We utilize entropy-regularized UOT with $\varepsilon = 0.01$ (Appendix B.1) and peform a small grid search over $\tau$. Note that while we train estimators to learn an unbalanced Monge map, i.e. Monge maps between the rescaled distribution $\tilde{\mu}$ and $\tilde{\nu}$, we evaluate them on balanced distributions $\mu$ and $\nu$.

### 5.1 SINGLE-CELL TRAJECTORY INFERENCE

OT has been successfully applied to model cell trajectories in the discrete setting (Schiebinger et al., 2019). The need for scalability motivates the use of neural Monge maps. Hence, we leverage OT-ICNN to model the evolution of cells on a single-cell RNA-seq dataset comprising measurements of the developing mouse pancreas at embryonic days 14.5 and 15.5 (Appendix F.5.1). To evaluate how accurately OT-ICNN recovers biological ground truth, we rely on the established single-cell trajectory inference tool CellRank (Lange et al., 2022).

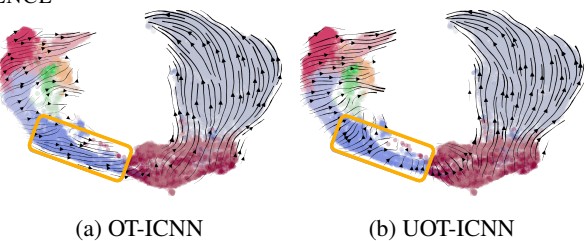

(a) OT-ICNN        (b) UOT-ICNN

Figure 3: Velocity stream embedding plots (Appendix F.5.6). The orange box highlights the direction of Ngn3 EP cells. With OT-ICNN these move to the "right", which contradicts biological ground truth. This is due to the distribution shift shown in Appendix F.5.1 and demonstrates the need to incorporate unbalancedness.

The developing pancreas can be divided into two major cell lineages, the endocrine branch (EB), and the non-endocrine branch (NEB) (Appendix F.5.4). It is known that once a cell has committed to either of these lineages, it won't develop into a cell belonging to the other lineage (Bastidas-Ponce

et al., 2019). The Ngn3 EP cluster is a population of early development in the EB, and hence cells belonging to this population are similar in their gene expression profile to cells in the NEB branch. Thus, when measuring the performance of OT-ICNN, we consider this cell population isolatedly. We evaluate OT-ICNN and UOT-ICNN by considering the predicted evolution of a cell.

We report the mean percentage of correct cell transitions for each lineage in Table 1, with full results in Appendix C.5. While UOT-ICNN consistently outperforms its balanced counterpart, the effect of the unbalanced neural Monge map estimator is particularly significant in the Ngn3 EP population. This improvement is visually confirmed in Figure 3. To demonstrate the biological relevance of our proposed method, we compare the results attained with OT-ICNN and UOT-ICNN with established trajectory inference methods in single-cell biology (Appendix C.5).

Table 1: Evaluation of unbalancedness in OT-ICNN based on correct cell type transitions.

| Model | Correct transitions | | |
|---|---|---|---|
| | EB | **Ngn3 EP** | NEB |
| OT-ICNN | 54% | 7% | 67% |
| UOT-ICNN | **57%** | **69%** | **85%** |

## 5.2 Modeling Perturbation Responses

In-silico prediction of cellular responses to drug perturbations is a promising approach to accelerate the development of drugs. Neural Monge maps have shown promising results in predicting the reaction of cells to different drugs (Bunne et al., 2021). Since Uscidda & Cuturi (2023) show that OT-MG outperforms OT-ICNN in this task, we continue this line of research by applying unbalanced OT-MG (UOT-MG) to predict cellular responses to 35 drugs from data profiled with 4i technology (Gut et al., 2018). We leverage distribution-level metrics to measure the performance of a map $\hat{T}$ modeling the effect of a drug. More precisely, given a metric $\Delta$, we keep for each drug a batch of unseen control cells $\mu_{\text{test}}$ and unseen treated cells $\nu_{\text{test}}$ and compute $\Delta(\hat{T}\sharp\mu_{\text{test}}, \nu_{\text{test}})$.

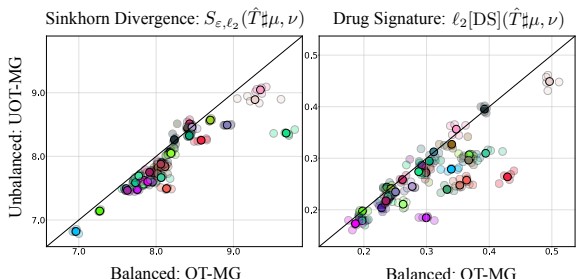

Figure 4: Fitting of a transport map $\hat{T}$ to predict the responses of cell populations to cancer treatments on 4i (upper plot), using balanced (OT-MG) and unbalanced Monge maps (UOT-MG) fitted with the Monge gap. A point below the diagonal indicates that unbalancedness improves performance.

We measure the predictive performances using the Sinkhorn divergence (Feydy et al., 2019) and the L2 drug signature (Bunne et al., 2021, Sec. 5.1). Figure 4 shows that adding unbalancedness through UOT-MG improves the performances for almost all 35 drugs, w.r.t. to both metrics. Each scatter plot displays points $z_i = (x_i, y_i)$ where $y_i$ is the performance obtained with UOT-MG and $x_i$ that of OT-MG, on a given treatment and for a given metric. A point below the diagonal $y = x$ refers to an experiment in which using an unbalanced Monge map improves performance. We assign a color to each treatment and plot five runs, along with their mean (the brighter point). For a given color, higher variability of points along the $x$ axis means that UOT-MG is more stable than OT-MG, and vice versa.

## 5.3 Unpaired Image Translation

**Metrics.** To evaluate the generative performance in the image translation settings, we employ the Fréchet inception distance (FID) (Heusel et al., 2018). In image translation, it is generally desirable to preserve certain attributes across the learned mapping. However, FID computed on the whole dataset does not take this into account. Hence, we also compute FID attribute-wise to evaluate how well relevant input features are preserved. Moreover, we consider the transport cost (T-Cost) $||\psi_1(\mathbf{x}_0) - \mathbf{x}_0||_2$ induced by the learned flow $\psi_1$. Full details are described in Appendix F.2.2.

**EMNIST.** To illustrate why unbalancedness can be crucial in unpaired image translation, we compare the performance of FM, OT-FM, and UOT-FM on the

Table 2: FID on EMNIST *digits → letters*

| Method | $0 \to O$ | $1 \to I$ | $8 \to B$ | Average |
|---|---|---|---|---|
| FM | 30.1 | 21.2 | 122.8 | 58.0 |
| OT-FM | **9.2** | 18.6 | 83.1 | 36.9 |
| UOT-FM | 12.5 | **13.9** | **42.6** | **23.0** |

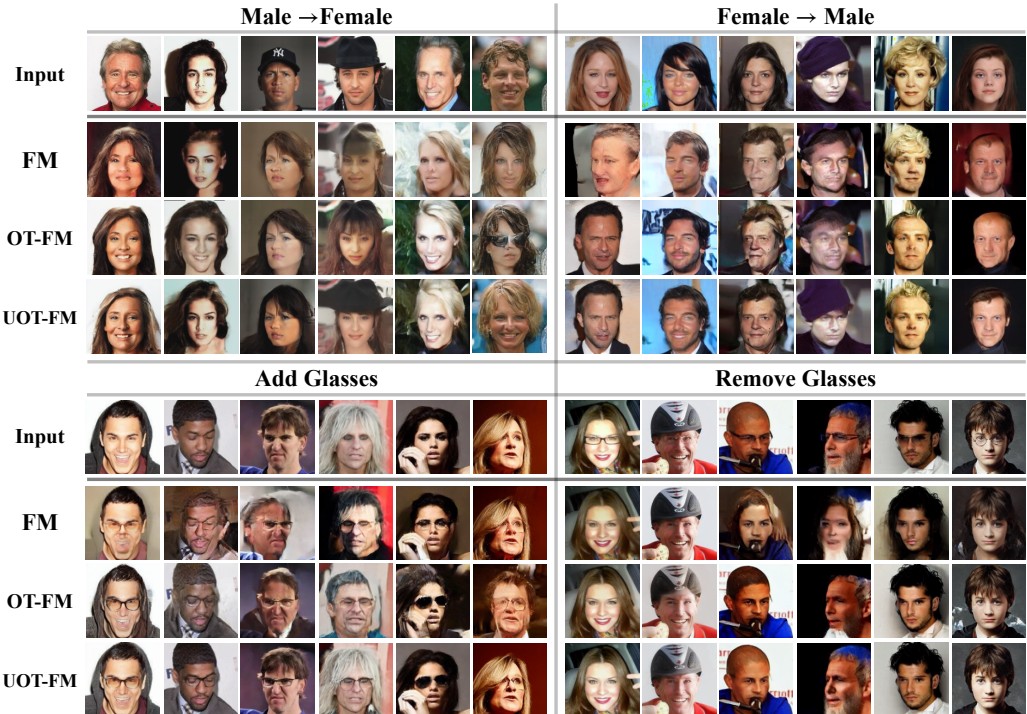

Figure 5: CelebA 256x256 translated test samples with FM, OT-FM, and UOT-FM.

EMNIST (Cohen et al., 2017) dataset translating the $digits$ $\{0, 1, 8\}$ to $letters$ $\{O, I, B\}$. We select this subset to obtain a desired class-wise mapping $\{0 \rightarrow O, 1 \rightarrow I, 8 \rightarrow B\}$. As seen in Appendix F.2.1 samples of class $B$ and $I$ are underrepresented in the target distribution. Because of this class imbalance, not all digits can be mapped to the correct corresponding class of letters. Figure 1 sketches how unbalanced Monge maps can alleviate this restriction. Indeed, Table 5 confirms this behavior numerically, while Figure 6 visually corroborates the results. The difference is especially striking for mapping class $8$ to $B$, where the largest distribution shift between source and target occurs. While this class imbalance is particularly noticeable in the considered task, these distribution-level discrepancies can be present in any image translation task, making unbalancedness crucial to obtain meaningful mappings with OT-FM.

**CelebA.** To demonstrate the applicability of our method on more complex tasks in computer vision, we benchmark on four translation tasks in the CelebA dataset (Liu et al., 2015), namely $Male \rightarrow Female$, $Female \rightarrow Male$, *Remove Glasses*, and *Add Glasses*. Here, the distribution shift occurs between image attributes. For example, when translating from $Male$ to $Female$, samples of males with hats (10.1%) significantly outnumber female ones (3.1%). Roughly speaking, this implies that when satisfying the mass conservation constraint, over half of the males with hats will be mapped to females without one. To compare to established methods, we use a common setup in high-dimensional image translation (Torbunov et al., 2022; Nizan & Tal, 2020; Zhao et al., 2021) where images are cropped and reshaped to 256x256. Due to computational reasons, we employ latent FM (Dao et al., 2023) leveraging a pretrained Stable Diffusion (Rombach et al., 2022) variational auto-encoder (VAE), which compresses images from $H \times W \times C$ to $\frac{H}{8} \times \frac{W}{8} \times 4$. We run FM, OT-FM, and UOT-FM in latent space, based on which we also compute the minibatch OT couplings. We compare UOT-FM against FM, OT-FM, CycleGAN (Zhu et al., 2017), and UVCGAN (Torbunov et al., 2022), which achieves current state-of-the-art results. While UOT-FM only requires training one network and the selection of one hyperparameter, established methods require more complex model and network choices as described in Appendix E.3.

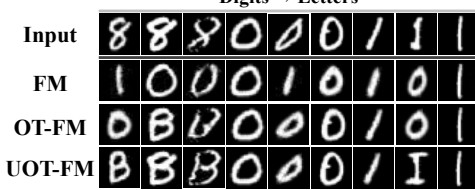

Figure 6: Samples from EMNIST translating $digits \rightarrow letters$ with different FM methods.

We report generative performance as well as transport cost results in Table 3. UOT-FM outperforms CycleGAN and OT-FM across all tasks with respect to the FID score while lowering the learned transport cost. Table 4 reports the attribute-wise FID between different FM methods. As expected,

Table 3: Comparison of results on CelebA 256x256 measured by FID and transport cost (T-Cost) over different tasks. Results marked with * are taken from Torbunov et al. (2022).

| Method | FID | T-Cost | FID | T-Cost | FID | T-Cost | FID | T-Cost |
|--------|-----|--------|-----|--------|-----|--------|-----|--------|
| | *Male → Female* | | *Female → Male* | | *Remove Glasses* | | *Add Glasses* | |
| UVCGAN | **9.6*** | - | **13.9*** | - | **14.4*** | - | **13.6*** | - |
| CycleGAN | 15.2* | - | 22.2* | - | 24.2* | - | 19.8* | - |
| FM | 17.5 | 89.2 | 18.6 | 91.2 | 25.6 | 74.0 | 30.5 | 73.9 |
| OT-FM | 14.6 | 59.3 | 14.8 | 59.7 | 20.9 | 47.1 | 20.1 | 45.7 |
| UOT-FM | 13.9 | **56.8** | 14.3 | **56.4** | 18.5 | **44.3** | 18.3 | **42.0** |

Table 4: CelebA 256x256 attribute-wise FID on different tasks for different FM methods.

| | *Male → Female* | | | | *Female → Male* | | | |
|--------|--------|-------|----------|-------------|--------|------|----------|-------------|
| Method | Glasses | Hat | Gray hair | **Average** | Glasses | Hat | Gray hair | **Average** |
| FM | 66.6 | 117.0 | 71.7 | 85.1 | 40.2 | 86.2 | 60.5 | 62.3 |
| OT-FM | 63.1 | 111.5 | 64.6 | 79.7 | 38.8 | 74.1 | 54.0 | 55.6 |
| UOT-FM | **62.6** | **108.5** | **62.7** | **77.9** | **38.2** | **72.4** | **51.1** | **53.9** |

| | *Remove Glasses* | | | | | *Add Glasses* | | | |
|--------|------|--------|------|-----------|-------------|------|--------|------|-----------|-------------|
| | Male | Female | Hat | Gray hair | **Average** | Male | Female | Hat | Gray hair | **Average** |
| FM | 46.9 | 47.7 | 110.6 | 89.1 | 74.3 | 19.6 | 43.2 | 46.6 | 39.7 | 37.3 |
| OT-FM | 42.3 | 40.3 | 99.5 | 83.1 | 66.3 | 12.0 | 36.9 | 44.4 | 27.4 | 30.2 |
| UOT-FM | **40.5** | **37.3** | **89.6** | **77.0** | **61.1** | **11.4** | **35.0** | **39.7** | **25.5** | **27.9** |

unbalancedness improves performance substantially. To confirm the superior performance of UOT-FM qualitatively, Figure 5 visualizes translated samples for FM, OT-FM, and UOT-FM. To demonstrate the strong performance of UOT holding in pixel space, we follow Korotin et al. (2022) and downsample images to 64x64 to evaluate FM, OT-FM, and UOT-FM translating $Male → Female$. We benchmark UOT against two principled approaches for unpaired image translation, namely NOT (Korotin et al., 2022) and CycleGAN. Appendix C.1 confirms the superiority of UOT-FM over OT-FM. These convincing results put UOT-FM forward as a new principled approach for unpaired image translation. Lastly, UOT-FM also improves performance with a low-cost solver and fewer function evaluations, which allows for higher translation quality on a compute budget (Appendix C.2).

**CIFAR-10 image generation.** The experiments above show that unbalanced neural Monge maps are mostly favorable over their balanced counterpart when translating between two data distributions. When translating from a source *noise*, i.e. a parameterized distribution that is easy to sample from, this task is referred to as *generative modeling*. Hence, we investigate whether UOT-FM also improves upon OT-FM in an image generation task. We benchmark on the CIFAR-10 (Krizhevsky et al.) dataset using the hyperparameters reported in Tong et al. (2023b) and show that also here UOT-FM improves upon OT-FM w.r.t. FID score. Moreover, we plot FID and transport cost convergence over training in Appendix C.7 and generated samples (C.8).

Table 5: OT-FM compared to UOT-FM with different $\tau$ on CIFAR-10 measure by FID and transport cost.

| Method | FID | T-Cost |
|--------|-----|--------|
| OT-FM | 3.59 | 104.53 |
| UOT-FM ($\tau = 0.99$) | 3.47 | 103.99 |
| UOT-FM ($\tau = 0.97$) | **3.42** | 102.98 |
| UOT-FM ($\tau = 0.95$) | 3.79 | **102.17** |

## 6 CONCLUSION

In this work, we present a new approach to incorporate unbalancedness into *any* neural Monge map estimator. We provide theoretical results that allow us to learn an unbalanced Monge map by showing that this is equivalent to learning the balanced Monge map between two rescaled distributions. This approach can be incorporated into *any* neural Monge map estimator. We demonstrate that unbalancedness enhances the performance in various unpaired domain translation tasks. In particular, it improves trajectory inference on time-resolved single-cell datasets and the prediction of perturbations on the cellular level. Moreover, we demonstrate that while OT-FM performs competitively in natural image translation, unbalancedness (UOT-FM) further elevates these results. While we use the squared Euclidean distance for all experiments, an interesting future direction lies in the exploration of more meaningful costs tailored specifically towards the geometry of the underlying data space.

## 7 REPRODUCIBILITY

For our new findings, we supply complete proofs in Appendix A. We detail the proposed algorithms in Appendix D and provide all implementation details for reproducing the results of all three tasks reported in this work in Appendix F. Additionally, the code to reproduce our experiments can be found at https://github.com/ExplainableML/uot-fm.

## 8 ACKNOWLEDGEMENTS

The authors would like to thank Alexey Dosovitskiy, Sören Becker, Alessandro Palma, Alejandro Tejada, and Phillip Weiler for insightful and encouraging discussions as well as their valuable feedback. Luca Eyring and Dominik Klein thank the European Laboratory for Learning and Intelligent Systems (ELLIS) PhD program for support. Zeynep Akata acknowledges partial funding by the ERC (853489 - DEXIM) and DFG (2064/1 – Project number 390727645) under Germany's Excellence Strategy. Fabian Theis acknowledges partial founding by the ERC (101054957 - DeepCell), consulting for Immunai Inc., Singularity Bio B.V., CytoReason Ltd, Cellarity, and an ownership interest in Dermagnostix GmbH and Cellarity. Views and opinions expressed are however those of the authors only and do not necessarily reflect those of the European Union or the European Research Council. Neither the European Union nor the granting authority can be held responsible for them.

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

# A  PROOFS

In this section, we prove Propositon 3.1, which we repeat here.

**Proposition 3.1** (Re-balancing the UOT problem)**.** *Let $\pi_{\mathrm{UOT}}$ be the solution of problem (UKP) between $\mu, \nu \in \mathcal{M}^+(\Omega)$, for $\tau_1, \tau_2 > 0$. Then, the following holds:*

1. *$\pi_{\mathrm{UOT}}$ solves the balanced problem (KP) between its marginal $\tilde{\mu} = p_1 \sharp \pi_{\mathrm{UOT}}$ and $\tilde{\nu} = p_2 \sharp \pi_{\mathrm{UOT}}$, which are re-weighted versions of $\mu$ an $\nu$ that have the same total mass. Indeed, $\tilde{\mu} = \bar{\phi}(-f^\star) \cdot \mu$ and $\tilde{\nu} = \bar{\phi}(-g^\star) \cdot \nu$, where $\bar{\phi} = (\phi^*)'$ and $f^\star, g^\star : \Omega \to \mathbb{R}$ are solution of (UDKP).*

2. *If we additionally assume that $\mu \ll \mathcal{L}_d$ and that the cost $c(\mathbf{x}, \mathbf{y}) = h(\mathbf{x} - \mathbf{y})$ with $h$ strictly convex, then $\pi_{\mathrm{UOT}}$ is unique and deterministic: $\pi_{\mathrm{UOT}} = (\mathrm{I}_d, T^\star)\sharp\tilde{\mu}$, where $T^\star = \mathrm{I}_d - \nabla h^* \circ f^\star$ is the Monge map between $\tilde{\mu}$ and $\tilde{\nu}$ for cost $c$.*

*Proof.* **Proof of point 1.** We remember that

$$\pi_{\mathrm{UOT}} \in \underset{\pi \in \mathcal{M}(\mathcal{X}, \mathcal{Y})}{\arg\min} \int_{\Omega \times \Omega} c(\mathbf{x}, \mathbf{y}) \, \mathrm{d}\pi(\mathbf{x}, \mathbf{y}) + \lambda_1 \mathrm{D}_\phi(p_1 \sharp \pi | \mu) + \lambda_2 \mathrm{D}_\phi(p_2 \sharp \pi | \mu) \tag{7}$$

so since we assume that $\phi'_\infty = +\infty$, the terms $\mathrm{D}_\phi(p_1 \sharp \pi | \mu)$ and $\mathrm{D}_\phi(p_2 \sharp \pi | \nu)$ are finite i.f.f. $p_1 \sharp \pi$ has a density w.r.t. $\mu$ and $p_2 \sharp \pi$ has a density w.r.t. $\nu$. Therefore, it exists $u, v : \mathbb{R}^d \to \mathbb{R}^+$ s.t. $\tilde{\mu} = p_1 \sharp \pi_{\mathrm{UOT}} = u \cdot \mu$ and $\tilde{\mu} = p_2 \sharp \pi_{\mathrm{UOT}} = v \cdot \nu$. Moreover, $\tilde{\mu}$ and $\tilde{\nu}$ have the same total mass, since by applying Fubini's Theorem twice, one has

$$\int_\Omega \mathrm{d}\pi_{\mathrm{UOT}}(\mathbf{x}, \mathbf{y}) = \int_\Omega \mathrm{d}\tilde{\mu}(\mathbf{x}) = \int_\Omega \mathrm{d}\tilde{\nu}(\mathbf{y}) \tag{8}$$

Moreover, from (Liero et al., 2018, Corollary 4.16), it exists $(f^\star, g^\star) \in \Phi_c(\mu, \nu)$ solution of the unbalanced Kantorovich dual (UDKP) between $\mu$ and $\nu$ s.t. $f \oplus g = c$, $\pi_{\mathrm{UOT}}$-a.e. This implies that $(f^\star, g^\star)$ solves the balanced Kantorovich dual problem (DKP) between $\tilde{\mu}$ and $\tilde{\nu}$. Indeed, since $\pi_{\mathrm{UOT}}$ has $\tilde{\mu}$ and $\tilde{\nu}$ as marginals, this equality yields

$$\int_\Omega f^\star(\mathbf{x}) \, \mathrm{d}\tilde{\mu}(\mathbf{x}) + \int_\Omega g^\star(\mathbf{y}) \, \mathrm{d}\tilde{\nu}(\mathbf{y}) = \int_\Omega c(\mathbf{x}, \mathbf{y}) \, \mathrm{d}\pi_{\mathrm{UOT}}(\mathbf{x}, \mathbf{y}) \, . \tag{9}$$

On the other hand, for any $(f, g) \in \Phi_c(\tilde{\mu}, \tilde{\nu})$, one has

$$\int_\Omega f(\mathbf{x}) \, \mathrm{d}\tilde{\mu}(\mathbf{x}) + \int_\Omega g(\mathbf{y}) \, \mathrm{d}\tilde{\nu}(\mathbf{y}) = \int_\Omega f \oplus g(\mathbf{x}, \mathbf{y}) \, \mathrm{d}\pi_{\mathrm{UOT}}(\mathbf{x}, \mathbf{y}) \leq \int_{\Omega \times \Omega} c(\mathbf{x}, \mathbf{y}) \, \mathrm{d}\pi_{\mathrm{UOT}}(\mathbf{x}, \mathbf{y}) \tag{10}$$

which provides

$$\sup_{(f,g) \in \Phi_c(\tilde{\mu}, \tilde{\nu})} \int_\Omega f(\mathbf{x}) \, \mathrm{d}\tilde{\mu}(\mathbf{x}) + \int_\Omega g(\mathbf{y}) \, \mathrm{d}\tilde{\nu}(\mathbf{y}) \leq \int_{\Omega \times \Omega} c(\mathbf{x}, \mathbf{y}) \, \mathrm{d}\pi_{\mathrm{UOT}}(\mathbf{x}, \mathbf{y}). \tag{11}$$

Additionally, since $(f^\star, g^\star) \in \Phi_c(\mu, \nu)$, one has $(f^\star, g^\star) \in \Phi_c(\tilde{\mu}, \tilde{\nu})$ because we have shown that $\tilde{\mu} \ll \mu$ and $\tilde{\nu} \ll \nu$. Therefore, they are optimal dual potentials. Then, since $c$ is continuous and $\Omega$ is compact, strong duality holds (Santambrogio, 2015, Theorem 1.46) and thus

$$\sup_{(f,g) \in \Phi_c(\tilde{\mu}, \tilde{\nu})} \int_\Omega f(\mathbf{x}) \, \mathrm{d}\tilde{\mu}(\mathbf{x}) + \int_\Omega g(\mathbf{y}) \, \mathrm{d}\tilde{\nu}(\mathbf{y}) = \inf_{\pi \in \Pi(\tilde{\mu}, \tilde{\nu})} \int_{\Omega \times \Omega} c(\mathbf{x}, \mathbf{y}) \, \mathrm{d}\pi(\mathbf{x}, \mathbf{y})$$
$$= \int_{\Omega \times \Omega} c(\mathbf{x}, \mathbf{y}) \, \mathrm{d}\pi_{\mathrm{UOT}}(\mathbf{x}, \mathbf{y}), \tag{12}$$

which yields the optimality of $\pi_{\mathrm{UOT}}$ in the balanced problem (KP) between $\tilde{\mu}$ and $\tilde{\nu}$.

To conclude the proof of the first point, we show that $u = \frac{\mathrm{d}\tilde{\mu}}{\mathrm{d}\mu} = \bar{\phi}(-f)$ and $u = \frac{\mathrm{d}\tilde{\nu}}{\mathrm{d}\nu} = \bar{\phi}(-g)$. (Liero et al., 2018, Corollary 4.16) also states that $f^\star = -\phi' \circ u$ $\mu$-a.e. and $g^\star = -\phi' \circ v$ $\nu$-a.e. Since $\phi$ is strictly convex, $\phi'$ is invertible and $(\phi')^{-1} = (\phi^*)' = \bar{\phi}$ (Santambrogio, 2015, Box 1.12), so the result follows.

**Proof of point 2.** First, given that $\mu \ll \mathcal{L}_d$, one has $\tilde{\mu} \ll \mathcal{L}_d$ since $\tilde{\mu} \ll \mu$. Then, since $c(\mathbf{x}, \mathbf{y}) = h(\mathbf{x} - \mathbf{y})$ where $h$ is strictly convex, we can apply (Santambrogio, 2015, Theorem 1.17) to state that

the Monge map between $\tilde{\mu}$ and $\tilde{\nu}$ exists and is unique, and $T^\star = \mathrm{I}_d - \nabla h^* \circ f^\star$ since we have shown that $(f^\star, g^\star)$ are optimal balanced dual potential between $\tilde{\mu}$ and $\tilde{\nu}$. Then, since $c$ is continuous, $\Omega$ is compact and $\tilde{\mu}$ is atomless, the Kantorovich and the Monge formulation coincide (Santambrogio, 2015, Theorem 1.33) so $\pi_{\mathrm{UOT}} = (\mathrm{I}_d, T^\star)\sharp\tilde{\mu}$ and it is unique.

$\square$

## B  COUPLING COMPUTATION

In this section, we lay out details w.r.t. the mini-batch coupling computation we leverage in our framework. In B.1 we detail entropy regularized OT, which we utilize as it offers a more efficient way of estimating couplings as opposed to computing the non-regularized one, which we lay out in B.3. Moreover, we discuss limitations that might arise given the coupling computation and in general with our proposed framework (B.2).

### B.1  ENTROPIC REGULARIZATION

When the measures $\mu$ and $\nu$ are instantiated as samples, as usual in a machine learning context, the Kantorovich problem (UKP) translates to a convex program whose objective function can be smoothed out using entropic regularization (Cuturi, 2013). For empirical measures $\hat{\mu}_n = \frac{1}{n}\sum_{i=1}^n \delta_{\mathbf{x}_i}$, $\hat{\nu}_n = \frac{1}{n}\sum_{j=1}^n \delta_{\mathbf{y}_j}$ and $\varepsilon > 0$, we form the cost matrix $\mathbf{C} = [c(\mathbf{x}_i, \mathbf{y}_j)]_{ij}$ and consider:

$$\min_{\mathbf{P} \geq 0} \ \langle \mathbf{P}, \mathbf{C} \rangle + \lambda_1 \mathrm{D}_\phi(\mathbf{P}\mathbf{1}_n | \tfrac{1}{n}\mathbf{1}_n) + \lambda_2 \mathrm{D}_\phi(\mathbf{P}^\top\mathbf{1}_n | \tfrac{1}{n}\mathbf{1}_n) - \varepsilon H(\mathbf{P}), \tag{13}$$

where $H(\mathbf{P}) = -\sum_{i,j=1}^n \mathbf{P}_{ij}\log(\mathbf{P}_{ij})$. With $\varepsilon \to 0$, we recover (UKP). Using $\varepsilon > 0$, the above Eq. (13) admits a tractable dual representation that can be leveraged to derive a fast computational procedure which generalizes the Sinkhorn algorithm and is commonly used in computational OT (Chizat et al., 2018b; Peyré & Cuturi, 2019; Séjourné et al., 2023). In practice, we follow Cuturi et al. (2022) and define $\tau_i = \frac{\lambda_i}{\lambda_i + \varepsilon}$. This facilitates hyper-parameter selection: $\tau_i \in (0, 1]$ and we recover the $i$-th hard marginal constraint with $\tau_i = 1$, when $\lambda_i \to +\infty$. In this work, we use entropic regularization with a small enough regularization strength $\varepsilon$ to approximate UOT couplings. In all the experiments, we use $\varepsilon = 0.01 \cdot \bar{\mathbf{C}}$, where $\bar{\mathbf{C}}$ denotes the mean of the cost matrix $\mathbf{C}$.

### B.2  LIMITATIONS

We consider the choice of the additional hyperparameter $\tau = (\tau_1, \tau_2)$ as the main challenge of our proposed framework. A way to make the choice of these hyperparameters independent of the scale of the data, and hence to some degree comparable across different tasks, is to scale the cost matrix of the discrete OT problem by its mean, which we leverage in all experiments as mentioned in Appendix B.1.

A second limitation is the implicit removal of outliers by UOT. This property is largely seen as a strength of UOT, e.g. Choi et al. (2023) show UOT's robustness to outliers. However, what is considered an outlier with respect to UOT is again dependent on the hyperparameter $\tau$. As one lowers $\tau$, the data pairs with lower distance attain more relative mass, and as $\tau \to 0$ sampling from the coupling approaches almost solely sampling the lowest distance pair. Depending on the application, there might be data points in the dataset that are "far" away from the remaining data points but are considered to be relevant to the model training. When choosing a lower $\tau$, these points might get removed by UOT. Hence, the choice of $\tau$ can be very important, but as discussed in Séjourné et al. (2023) the optimal choice is not evident and thereby usually facilitates at least a small grid search. Lastly, our framework also adds computational complexity, although negligible in most cases, as discussed in B.3.

### B.3  COMPUTATIONAL COMPLEXITY

We compute all mini-batch UOT couplings using entropic regularization, which allows the use of a generalization of Sinkhorn's algorithm (Chizat et al., 2018b), having the same $\mathcal{O}(n^2)$ time complexity, where $n$ depicts the batch size. The added computational overhead of our framework compared to

using the balanced alternative depends on the estimator used. We distinguish between two types of estimators in the following.

**Estimators that don't leverage OT couplings (e.g. OT-MG or OT-ICNNs).** These estimators require samples of the marginal distributions between which the Monge map is to be calculated. Thus, each of the training iterations for these models involves first calculating an unbalanced coupling $\hat{\pi}_{\text{UOT}}$ using samples $\mathbf{x}_1, \ldots, \mathbf{x}_n \sim \mu$ and $\mathbf{y}_1, \ldots, \mathbf{y}_n \sim \nu$, then evaluate their respective training losses $\mathcal{L}_{\text{OT-MG}}$ or $\mathcal{L}_{\text{OT-ICNN}}$ on samples $\tilde{\mathbf{x}}_1, \ldots, \tilde{\mathbf{x}}_n \sim \tilde{\mu}_n = p_1 \sharp \hat{\pi}_{\text{UOT}}$ and $\tilde{\mathbf{y}}_1, \ldots, \tilde{\mathbf{y}}_n \sim \tilde{\nu}_n = p_1 \sharp \hat{\pi}_{\text{UOT}}$. Thus, for both these estimators, the additional computational cost lies in calculating $\hat{\pi}_{\text{UOT}}$.

**Estimators that already leverage OT couplings (like OT-FM).** Each iteration of OT-FM consists of sampling and calculating a coupling $\hat{\pi}_{\text{OT}}$ from samples $\mathbf{x}_1, \ldots, \mathbf{x}_n \sim \mu$ and $\mathbf{y}_1, \ldots, \mathbf{y}_n \sim \nu$, then calculating the Flow Matching loss on samples of $\hat{\pi}_{\text{OT}}$. For UOT-FM, we directly replace the calculation of a balanced plan $\hat{\pi}_{\text{OT}}$ by an unbalanced plan $\hat{\pi}_{\text{OT}}$ from samples $\mathbf{x}_1, \ldots, \mathbf{x}_n \sim \mu$ and $\mathbf{y}_1, \ldots, \mathbf{y}_n \sim \nu$, then calculate the Flow Matching loss on $\hat{\pi}_{\text{OT}}$ samples in the same way. Since $\hat{\pi}_{\text{OT}}$ and $\hat{\pi}_{\text{UOT}}$ can be calculated with the same $\mathcal{O}(n^2)$ runtime complexity, using Sinkhorn's algorithm (Cuturi, 2013) and its unbalanced generalization (Chizat et al., 2018b), UOT-FM has exactly the same runtime complexity as OT-FM. Therefore, since FM operates at a coupling level, incorporating unbalancedness via UOT-FM does not add any additional computational cost.

In the first case, the relative computational overhead depends on the cost of a gradient step. In Pooladian et al. (2023a); Tong et al. (2023b) it was shown that even coupling computations of cost $\mathcal{O}(n^3)$ are negligible compared to the gradient step of the Flow Matching loss. In this work, which involves computations scaling quadratically $\mathcal{O}(n^2)$ rather than cubically, this observation applies even more strongly. Hence, in most settings, our framework will not significantly impact training time.

# C  ADDITIONAL EMPIRICAL RESULTS

In this section, we report additional results: In C.1 results in pixel space for 64x64 CelebA translating $Male \to Female$. Then for 256x256 CelebA translating $Male \to Female$ we evaluate the performance of UOT-FM compared to OT-FM with few function evaluations and a low-cost Euler solver (C.2) and in C.3 we report results for different levels of unbalancedness $\tau$ in UOT-FM. Moreover, we compare the performance of OT-ICNN and UOT-ICNN to established methods in single-cell trajectory inference (C.5). Furthermore, we benchmark UOT-ICNN against competing unbalancedness methods proposed in Lübeck et al. (2022) and Yang & Uhler (2019) on sciPlex perturbation dataset (Srivatsan et al., 2020) in C.6. Lastly, we plot FID and transport cost over training (C.7) for the generative modeling experiment on CIFAR-10 as well as some randomly generated samples in C.8.

## C.1  CELEBA IN PIXEL-SPACE

Table 6: CelebA 64x64 *Male → Female*. Results denoted with ∗ are taken from Korotin et al. (2022).

| Method | FID | T-Cost | Glasses | Hat | Gray hair | Average |
|---|---|---|---|---|---|---|
| | | | *attribute-wise FID* | | | |
| NOT | 13.23* | - | - | - | - | - |
| CycleGAN | 17.74* | - | - | - | - | - |
| OT-FM | 11.52 | 40.37 | 47.40 | 85.80 | 49.53 | 60.91 |
| UOT-FM | **11.09** | **37.38** | **47.13** | **84.29** | **46.73** | **59.38** |

## C.2 CELEBA WITH A LOW-COST SOLVER

Table 7: CelebA 256x256 *Male → Female* with a low-cost Euler solver and a varying number of function evaluations.

| | FID | | T-Cost | |
|---|---|---|---|---|
| **NFE** | OT-FM | UOT-FM | OT-FM | UOT-FM |
| Adaptive | 14.58 | **13.94** | 59.31 | **56.78** |
| 40 | 14.53 | **14.02** | 57.00 | **54.84** |
| 20 | 14.91 | **14.66** | 54.44 | **53.40** |
| 12 | 16.40 | **16.08** | 53.69 | **51.82** |
| 8 | 19.56 | **19.32** | 52.46 | **50.49** |
| 6 | 24.14 | **24.08** | 51.77 | **46.66** |
| 4 | 38.02 | **37.38** | 51.25 | **49.02** |
| 2 | 88.47 | **87.85** | 55.08 | **53.22** |

## C.3 CELEBA OVER DIFFERENT LEVELS OF UNBALANCEDNESS

Table 8: 256x256 CelebA *male → female* results for UOT-FM with different $\tau = \tau_1 = \tau_2$.

| | FID | T-Cost | FID Average |
|---|---|---|---|
| OT-FM | 14.58 | 59.31 | 79.7 |
| $\tau = 0.99$ | 14.75 | 58.94 | 80.3 |
| $\tau = 0.98$ | 14.23 | 57.63 | 79.8 |
| $\tau = 0.95$ | **13.94** | 56.78 | **77.9** |
| $\tau = 0.90$ | 14.30 | **55.24** | 78.8 |

## C.4 EFFECT OF BATCH-SIZE AND EPSILON ON EMNIST

Here, we empirically evaluate the effect of changing the batch size and $\varepsilon$ in UOT-FM applied to the EMNIST dataset translating $digits → letters$. We report the learned Transport Cost and attribute-wise average FID score as detailed in Appendix F.2.2. The results reported in Section 5.3 are obtained with a batch size of 256 and $\varepsilon = 0.01$. Figure 7a shows the effect of changing the batch size. Our results align with previous studies (Pooladian et al., 2023a; Tong et al., 2023b), which show that already a fairly low batch size yields competitive results with OT mini-batch couplings, and increasing it over 128 does not change results significantly. Only when we decrease the batch size below 32, do we observe a significant drop in performance. For all batch size results, we keep the number of samples seen during training the same set to $256 \cdot 500k$.

Secondly, Figure 7b confirms empirically how the performance of UOT-FM approaches FM as we increase $\varepsilon$. Additionally, we find that lowering $\varepsilon$ further than $0.01$ does not improve the learned transport cost and average FID significantly while increasing the runtime of the coupling computation.

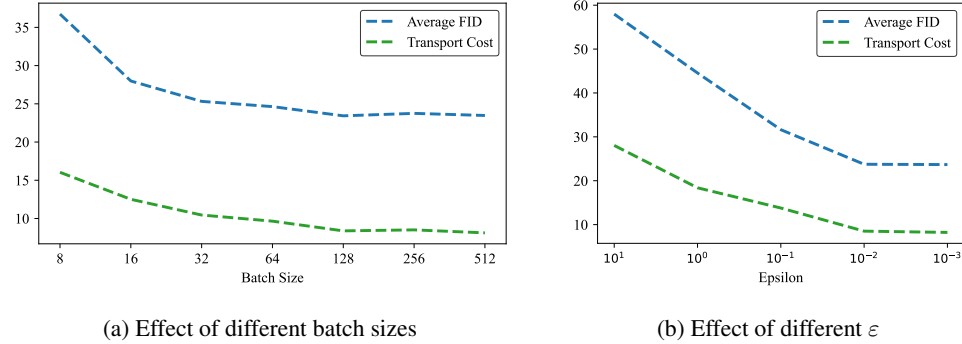

(a) Effect of different batch sizes          (b) Effect of different $\varepsilon$

Figure 7: Results on EMNIST with UOT-FM translating $digits \rightarrow letters$.

## C.5 SINGLE-CELL TRAJECTORY INFERENCE COMPARISON TO ESTABLISHED METHODS

We compare OT-ICNN and UOT-ICNN against established methods in single-cell trajectory inference. We benchmark against scVelo (Bergen et al., 2019), TrajectoryNet (Tong et al., 2020), and Waddington OT (WOT) (Schiebinger et al., 2019). Table 9 summarizes results, while the full transition probabilities are reported in Table 10. UOT-ICNN outperforms all competing methods on the EB and NEB branches while performing second best on the Ngn3 EP cells. Additionally, UOT-ICNN improves or keeps equal performance upon OT-ICNN in ten out of thirteen cell transitions. Implementation details of each competing method are described in Appendix F.5.5.

Table 9: Evaluation of different trajectory inference methods based on correct cell type transitions.

| Model | Correct transitions | | |
|---|---|---|---|
| | EB | Ngn3 EP | NEB |
| TrajectoryNet | 0.33 | 0.01 | 0.71 |
| scVelo | 0.44 | **0.99** | 0.39 |
| WOT | 0.45 | 0.50 | 0.72 |
| OT-ICNN | 0.53 | 0.07 | 0.67 |
| UOT-ICNN | **0.57** | 0.69 | **0.85** |

Table 10: Cell type transition probabilities between cell types $A$ and $B$ such that cell type $A$ maps exclusively to cell type $B$. For each column, we underline the best, second best, and third best methods.

| Model | FA → A | A → A | FB → B | B → B | FD → D | D → D | FE → E | E → E | NE → ED | NL → ED | DU → DU | T → AC | AC → AC |
|---|---|---|---|---|---|---|---|---|---|---|---|---|---|
| TrajectoryNet | 0.07 | 0.46 | 0.07 | 0.39 | 0.11 | **0.75** | 0.10 | 0.52 | 0.00 | 0.01 | **0.32** | **0.79** | **0.99** |
| scVelo | **0.79** | **0.80** | **0.30** | **0.61** | 0.03 | 0.52 | 0.04 | **0.56** | **0.98** | **1.00** | 0.32 | 0.04 | 0.90 |
| WOT | **0.37** | **0.62** | **0.18** | 0.44 | **0.19** | 0.74 | **0.55** | 0.49 | **0.50** | **0.50** | **0.82** | 0.48 | 0.84 |
| OT-ICNN | **0.51** | 0.54 | 0.11 | **0.70** | **0.39** | **0.94** | **0.48** | **0.62** | 0.07 | 0.08 | 0.01 | **1.00** | **1.00** |
| UOT-ICNN | 0.25 | **0.60** | **0.29** | **0.75** | **0.78** | **0.99** | 0.36 | **0.55** | **0.65** | **0.73** | **0.59** | **1.00** | **1.00** |

## C.6 BENCHMARK AGAINST COMPETING UNBALANCEDNESS ESTIMATORS

To assess the performance of our proposed approach UOT-ICNN, we benchmark it against competing methods proposed in Lübeck et al. (2022) and Yang & Uhler (2019). While our proposed approach for unbalanced neural Monge maps is significantly different from Lübeck et al. (2022), both approaches build upon the solver suggested in Makkuva et al. (2020). We use the exact same architecture and training procedure for both approaches and only perform the resampling in a different way. In detail, UOT-ICNN solves one discrete unbalanced OT problem between a batch of the source distribution and a batch of the target distribution. In contrast, Lübeck et al. (2022) compute an unbalanced discrete OT solution between the push-forward of the source distribution and the target distribution to estimate the left rescaling factor. Moreover, they compute a second discrete OT solution between the pull-back of the target batch and the source batch to obtain an estimate of the right rescaling factor. We also compare to Yang & Uhler (2019) but would like to highlight that we can only approximate a similar setup, e.g. by also training for 1000 iterations.

We evaluate the models on the sciPlex pertubation data (Srivatsan et al., 2020). We chose drugs that were reported to have a strong signal. The data was downloaded from `https://github.com/bunnech/cellot/tree/main`. We compute a 30-dimensional PCA embedding and evaluate the performance with the Sinkhorn divergence (Feydy et al., 2019) (with entropy regularization parameter $\varepsilon = 0.01$) (Feydy et al., 2019), analogously to the experimental setup for comparing OT-MG with UOT-MG in Section 5.2. Then we report the mean Sinkhorn divergence across all drugs for different levels of unbalancedness $\tau$ in Table 11. UOT-ICNN outperforms competing methods across all $\tau$. In particular, Lübeck et al. (2022) achieves close performance to UOT-ICNN with $\tau = 0.95$ but seems to diverge for lower values of $\tau$.

Table 11: Sinkhorn divergence between predicted target and target distribution on sciPlex data per model. We highlight **best** and second best performance.

| Method | Sinkhorn Divergence | | | | | |
| --- | --- | --- | --- | --- | --- | --- |
| | $\tau = 0.95$ | $\tau = 0.9$ | $\tau = 0.85$ | $\tau = 0.8$ | $\tau = 0.75$ | $\tau = 0.7$ |
| UOT-ICNN | **25.95** | **28.05** | **29.60** | **30.68** | **31.42** | **32.15** |
| Lübeck et al. (2022) | 26.39 | 1286.48 | 1841.31 | 3758.46 | 12913.06 | 27031.94 |
| Yang & Uhler (2019) | 37.40 | 37.79 | 37.84 | 38.03 | 38.09 | 37.83 |

## C.7 CIFAR-10 CONVERGENCE OVER TRAINING

Here, we plot FID score and transport cost over training. As expected introducing more unbalancedness by lowering $\tau$ reduces the learned transport cost. Generative performance increases with the introduction of unbalancedness offering faster convergence up until a certain level.

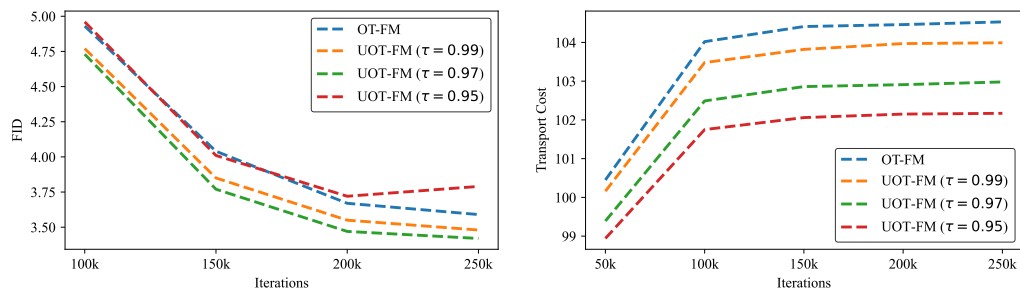

(a) FID score over training iterations.  (b) Learned transport cost over training iterations.

Figure 8: Convergence of training on CIFAR-10 plotted for OT-FM and UOT-FM with different $\tau$.

## C.8 CIFAR-10 Generated Samples

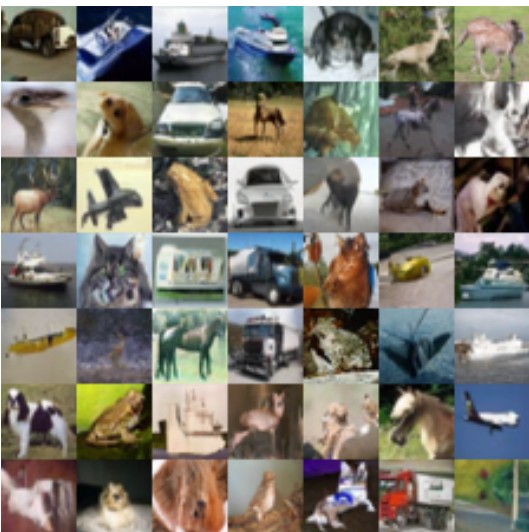

Figure 9: Randomly generated images with UOT-FM ($\tau = 0.97$) trained on CIFAR-10.

## D Algorithms

In this section, we describe the full algorithms for **[i]** estimating unbalanced Monge maps with *any* Monge map estimator in Algorithm 1, **[ii]** computing unbalanced optimal transport Flow Matching in Algorithm 2.

---

**Algorithm 1:** Neural Unbalanced Monge maps

---

**Require:** Source and target distribution $\mu, \nu$, unbalancedness parameters $\tau = (\tau_1, \tau_2)$, loss function of a balanced Monge map estimator $\mathcal{L}_{\text{OT}}$, solver $\text{Solver}_\tau$, batch size $n$, number of iterations $n_{\text{iter}}$, boolean flag learn_rescaling whether to learn the re-weightings functions, neural re-weighting functions $u_\theta, v_\theta$

1: **for** $l = 1, \ldots, n_{\text{iter}}$ **do**
2:      Sample batches $\mathbf{x}_1 \ldots \mathbf{x}_n \sim \mu$ and $\mathbf{y}_1 \ldots \mathbf{y}_n \sim \nu$
3:      Compute coupling $\hat{\pi}_{\text{UOT}} \leftarrow \text{Solver}_\tau(\{\mathbf{x}_i\}_{i=1}^n, \{\mathbf{y}_j\}_{i=1}^n)$
4:      Sample new batches $(\{\tilde{\mathbf{x}}_i\}_{i=1}^n, \{\tilde{\mathbf{y}}_j\}_{i=1}^n) \sim \hat{\pi}_{\text{UOT}}$
5:      Compute loss $\mathcal{L}_{\text{UOT}}(\theta) \leftarrow \mathcal{L}_{\text{OT}}(\theta; \{\tilde{\mathbf{x}}_i\}_{i=1}^n, \{\tilde{\mathbf{y}}_j\}_{i=1}^n)$
6:      **if** learn_rescaling **then**
7:          $\mathbf{a} \leftarrow \hat{\pi}_{\text{UOT}} \mathbf{1}_n$ and $\mathbf{b} \leftarrow \hat{\pi}_{\text{UOT}}^\top \mathbf{1}_n$
8:          $\mathcal{L}_{\text{UOT}}(\theta) \leftarrow \mathcal{L}_{\text{UOT}}(\theta) + \frac{1}{n} \sum_{i=1}^n (u_\theta(\mathbf{x}_i) - n \cdot a_i)^2 + \frac{1}{n} \sum_{j=1}^n (v_\theta(\mathbf{y}_j) - n \cdot b_j)^2$
9:      Update $\theta$ with $\nabla_\theta \mathcal{L}_{\text{UOT}}(\theta)$
10: **return** $\theta$

---

**Algorithm 2:** Unbalanced Optimal Transport Flow Matching (UOT-FM)

---

**Require:** Source and target distributions $\mu, \nu$, unbalancedness parameters $\tau_1, \tau_2$, parameterized vector field $v_\theta$.

1: **while** Training **do**
2:      Sample batches $\boldsymbol{x}_0 \sim \mu, \boldsymbol{x}_1 \sim \nu, t \sim \mathcal{U}(0, 1)$
3:      Compute coupling $\hat{\pi}_{\text{UOT}} = \text{UOT}(\boldsymbol{x}_0, \boldsymbol{x}_1, \tau_1, \tau_2)$
4:      Sample new batches $(\tilde{\boldsymbol{x}}_0, \tilde{\boldsymbol{x}}_1) \sim \hat{\pi}_{\text{UOT}}$
5:      Compute flow $\tilde{\boldsymbol{x}}_t = (1-t)\tilde{\boldsymbol{x}}_0 + t\tilde{\boldsymbol{x}}_1$
6:      Compute loss $\mathcal{L}_{\text{FM}}(\theta) = ||v_\theta(t, \tilde{\boldsymbol{x}}_t) - u_t(\tilde{\boldsymbol{x}}_t|\tilde{\boldsymbol{x}}_1)||^2$
7:      Update $\theta$ with $\nabla_\theta \mathcal{L}_{\text{FM}}(\theta)$
8: **return** $v_\theta$

---

# E    RELATED WORK FOR TRANSLATION TASKS

Here, we describe related work specific to the three different domain translation tasks.

## E.1    RELATED WORK IN SINGLE-CELL TRAJECTORY INFERENCE

Optimal transport has been established as a trajectory inference method in single-cell genomics, spearheaded by Schiebinger et al. (2019). The high computational burden of discrete OT has led to the development of low-rank solvers, which have been studied in the context of single-cell trajectory inference methods in Klein et al. (2023). Proofs of concept have also been conducted by Tong et al. (2020; 2023b;a). For developmental single-cell data, OT-based methods have been established as competitive methods in the field (Lange et al., 2022).

## E.2    RELATED WORK IN PERTURBATIONAL PREDICTIONS

Studying the effect of different perturbations on a single-cell level is a recent field in single-cell biology enabled by the progress of machine learning (Ji et al., 2021). First models were based on VAEs (Ji et al., 2021), while recently, the community has focused on OT-based methods (Bunne et al., 2021; Lübeck et al., 2022; Uscidda & Cuturi, 2023).

## E.3    RELATED WORK IN UNPAIRED IMAGE TRANSLATION

One of the seminal works in unpaired image translation was CycleGAN (Zhu et al., 2017), which introduced a cycle-consistency loss to encourage translations that can reconstruct the original image when run through a forward-backward cycle between domains. Several state-of-the-art methods have since built upon CycleGAN's foundational approach, seeking to improve the realism and variability of generated images. Notable examples include STARGAN (Choi et al., 2018), which extended CycleGAN to handle multiple attributes simultaneously, as well as CUT (Park et al., 2020), and most recently UVCGAN (Torbunov et al., 2022), which employs self-supervised pre-training and an optimized architecture, a UNet Vision Transformer (Dosovitskiy et al., 2021). Note, that UOT-FM trains one network with one loss function and one hyperparameter $\tau$ while in contrast, CycleGAN-based approaches train at least four networks with four loss functions and hyperparameters.

Diffusion models (DMs) (Song et al., 2020) have achieved ground-breaking results in image generation. However, DMs were designed with only a Gaussian noise source distribution in mind and thus are not directly applicable to image translation. This sparked the development of the extension of DMs to the unpaired image translation task. UNIT-DDPM (Sasaki et al., 2021) extends the CycleGAN concept to DMs by leveraging two DMs between domains with a cycle-consistency loss. ILVR (Choi et al., 2021) and SDEdit (Meng et al., 2022) rely on a test source image during inference without leveraging information from the training source distribution. EDGSE (Zhao et al., 2022) suggests alleviating this constraint by pre-training two energy functions on the respective source and target domain. Very recently, a different approach was proposed based on the framework of Diffusion Schrödinger Bridges (Bortoli et al., 2023) as applied in Liu et al. (2023).

In contrast to DMs, OT-FM offers a more natural framework for unpaired image translation out of the box. It has been applied to CelebA encoded in a 128-dimensional VAE embedding (Tong et al., 2023b), but without comparison to any established methods and without reporting common metrics in image translation like FID. However, they show that OT-FM improves performance upon FM with the independent coupling, which we confirm with our results also holds in higher dimensional embeddings (Section 5.3) and pixel-space (Appendix C.1). Rectified flow (Liu et al., 2022) can be interpreted as a version of FM that approximates OT-FM. It has been applied to high-quality image translation and reports results visually, but again without comparison to established methods and no numerical results.

# F    EXPERIMENTAL AND EVALUATION DETAILS

Our code is based on Jax (Bradbury et al., 2018) while utilizing parts of the DeepMind Jax ecosystem (Babuschkin et al., 2020).

### F.1 Unbalanced Coupling Algorithm

To compute the entropic unbalanced coupling we leverage the `ott-jax` library (Cuturi et al., 2022) with the Sinkhorn algorithm (Cuturi, 2013). We choose the entropic coupling with a small $\epsilon$ due to its computational benefits.

Table 12: Hyperparameters for unpaired image translation on CelebA.

|  | CelebA-256 gender | CelebA-256 glasses | CelebA-64 |
| --- | --- | --- | --- |
| Channels | 128 | 128 | 192 |
| ResNet blocks | 4 | 4 | 3 |
| Channels multiple | 2, 2, 2 | 2, 2, 2 | 1, 2, 3, 4 |
| Heads | 1 | 1 | 4 |
| Heads channels | 64 | 64 | 64 |
| Attention resolution | 16 | 16 | 32, 16, 8 |
| Dropout | 0.1 | 0.1 | 0.1 |
| GPU batch size | 64 | 64 | 32 |
| Effective Batch size | 256 | 256 | 256 |
| Iterations | 400k | 100k | 400k |
| Learning rate | 1e-4 | 1e-4 | 1e-4 |
| Scheduler | constant | constant | constant |
| EMA-decay | 0.9999 | 0.9999 | 0.9999 |

### F.2 Unpaired Image translation

For each of the CelebA image translation tasks, we use a very similar hyperparameter setup based upon the UNet architecture used in Dhariwal & Nichol (2021), where all FM models were trained with the exact same setup. We report these in Table 12. For EMNIST we leverage the MLPMixer architecture (Tolstikhin et al., 2021) with hyperparameters detailed in Table 13. Additionally, we use the Adam optimizer (Kingma & Ba, 2014) with $\beta_1 = 0.9, \beta_2 = 0.999, \epsilon = 1e-8$ and no weight decay for all image translation experiments.

During inference, we solve for $p_t$ at $t = 1$ using the adaptive step-size solver `Tsit5` with `atol=rtol=1e-5` implemented in the `diffrax` library (Kidger, 2021). For $\tau$ we employ a small grid search on the $Male \rightarrow Female$ task, where results are reported in Appendix C.3. $\tau = 0.95$ achieves the best performance and subsequently, we choose $\tau = 0.95$ for all other tasks, including in pixel-space. For the EMNIST experiment, we choose $\tau_1 = 0.9, \tau_2 = 1.0$.

#### F.2.1 Datasets

For all image translation experiments, we rescale images from $[0, 255]$ to $[-1.0, 1.0]$. In practice, it has been shown that data augmentation and regularization, e.g. random rotations or added noise, can be beneficial to achieve better generalization. We note that in our case, this entails learning a Monge map between adjusted distributions $\bar{\mu}$ and $\bar{\nu}$ instead of $\mu$ and $\nu$. In pratice, we regularize source and

Table 13: Hyperparameters for the MLPMixer used for the EMNIST experiments.

| | |
| --- | --- |
| Channels | 64 |
| Patch size | 4 |
| Token Mixer channels | 512 |
| Channel Mixer channels | 512 |
| Depth | 4 |
| Batch size | 256 |
| Iterations | 500k |
| Learning rate | 3e-4 |
| Scheduler | constant |
| EMA-decay | 0.9999 |

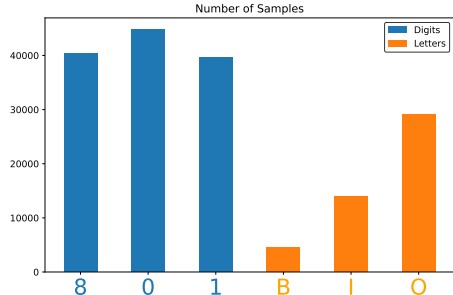

Figure 10: EMNIST data distribution.

target distribution with a small amount $\sigma$ of Gaussian noise such that we learn the Monge map between $\bar{\mu} = \mu * \mathcal{N}(0, \sigma^2 \boldsymbol{I})$ and $\bar{\nu} = \nu * \mathcal{N}(0, \sigma^2 \boldsymbol{I})$, where $*$ denotes the convolution operator. In the FM framework, this is equivalent to considering Gaussian probability paths $p_t(\mathbf{x}|\mathbf{x}_t) = \mathcal{N}(\mathbf{x}|\mathbf{x}_t, \sigma^2 \boldsymbol{I})$, which is implemented by adding a small amount of noise to the computation of $\boldsymbol{x}_t$ such that $\boldsymbol{x}_t = (1-t)\boldsymbol{x}_0 + t\boldsymbol{x}_1 + \sigma\epsilon$, where $\epsilon \sim \mathcal{N}(0, \boldsymbol{I})$.

**EMNIST.** We leverage a subset of EMNIST, where we take $digits$ $\{0, 1, 8\}$ as source and $letters$ $\{O, I, B\}$ as target distribution. The grayscale images are of shape 28x28. The resulting translation task contains a large distribution shift as visualized in Figure 10.

**CelebA.** CelebA contains images of size 218x178. Following Torbunov et al. (2022); Nizan & Tal (2020); Zhao et al. (2021) we do not use the validation dataset for training, but instead add it to the test dataset. Then, the gender swap task contains about 68k males and 95k females while the glasses task entails 11k samples with and 152k without glasses. For CelebA-256 we again follow a similar setup to Torbunov et al. (2022) and upsize images to 313x256. Then instead of random cropping we center crop all images to 256x256. This is done to pre-compute all embeddings for the Stable Diffusion VAE. Note, that this is a disadvantage for the benchmarked FM methods as they cannot utilize the benefit of random cropping during training like Torbunov et al. (2022). For Celeba-64 we resize all images to 64x64.

### F.2.2 METRIC COMPUTATION DETAILS

For general FID computation, we follow Torbunov et al. (2022); Korotin et al. (2022) and compare translated test samples from the source distribution against the statistic from the test target distribution. For the attribute-wise computation, the number of test samples can be very low depending on the task and attribute. Thus, we compute the attribute-wise FID comparing translated test samples to the statistics of the whole dataset w.r.t. the given attribute. For all FID computations, we use the `jax-fid` library. The transport cost $\|\psi_1(\mathbf{x}_0) - \mathbf{x}_0\|_2$ is always computed and averaged over the hold-out test set. Note that we compute transport cost in pixel space, scaled back to $[0, 255]$. The transport cost between the rescaled measures $\mathrm{W}_c(\tilde{\mu}, \tilde{\nu})$ is always less than or equal to the non-rescaled one $\mathrm{W}_c(\mu, \nu)$. With this metric, we aim to measure whether this also generalizes to unseen samples from the balanced source distribution $\boldsymbol{x}_0 \sim \mu$ when training UOT-FM.

### F.3 IMAGE GENERATION

For the CIFAR-10 experiments, we follow almost the exact setup reported in Tong et al. (2023b) except that we compute mini-batch couplings with entropy regularized OT as detailed in Appendix B.1. We report results at iteration $250k$ reproducing the OT-FM results from Tong et al. (2023b).

### F.4 SINGLE-CELL PERTURBATION RESPONSES

For each drug, we model the neural map $T_\theta$ using an MLP with hidden layer size $[\max(128, 2d), \max(128, 2d), \max(64, d), \max(64, d)]$ where $d$ is the dimension of the data. We train it for $n_{\text{iter}} = 10,000$ iterations with a batch size $n = 1,024$ and the Adam optimizer (Kingma & Ba, 2014) using a learning rate $\eta = 0.001$, along with a polynomial schedule of power $p = 1.5$ to

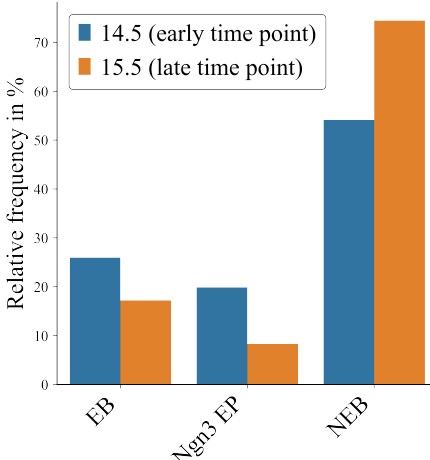

Figure 11: Distribution over the selected lineages and time points. Cells of the non-endocrine branch are much more abundant at the later time point due to very high proliferation rates of Acinar and Ductal cells.

decrease it to $10^{-5}$. Finally, we select the unbalancedness parameters using a grid search, imposing $\tau_1 = \tau_2$ and selecting among three values $\tau_i \in \{0.99, 0.95, 0.9\}$.

### F.5 SINGLE-CELL TRAJECTORY INFERENCE

#### F.5.1 PANCREATIC ENDOCRINOGENESIS DATA

The pancreatic endocrinogenesis data includes samples of embryonic days 14.5 and 15.5 (Bastidas-Ponce et al., 2019). After standard preprocessing $16,206$ genes remained. The OT-ICNN-based algorithms and Waddington OT were run on the 50 principal components. Figure 11 visualizes the distribution shift between the two time points. Note the increase in the NEB branch, which causes Ngn3 EP cells to be mapped to NEB without accounting for unbalancedness in OT-ICNN.

#### F.5.2 MODEL ARCHITECTURE

An Input Convex Neural Network (ICNN) parameterizes a function $f$ such that $f$ is convex with respect to its input by imposing certain constraints (Amos et al., 2017). Following Makkuva et al. we train two ICNNs, denoted by $f$ and $g$, with the following architecture (Makkuva et al., 2020):

- $K$ *dense* layers consisting of weights $A_0, \ldots, A_K$ applied to the raw input $x$,
- $K-1$ *positive dense* layers consisting of non-negative weights $W_1, \ldots, W_K$ applied to intermediate outputs $z_{k-1}$ as defined below.

Then, layer $k$ is defined as

$$z_k = \phi((W_k z_{k-1}) + (A_k x + b_k)) \tag{14}$$

where $\phi$ is a convex non-decreasing activation function, $b_k$ the bias term and $A_k$ the weight matrix. In the last layer, we apply no activation function. Additionally, we use a quadratic first layer:

$$z_0 = (\phi(A_0 x + b_0))^2 \tag{15}$$

We enforce the non-negativity constraint on of the weights $W$ by weight clipping, while we only use a penalization term for negative weights of $g$

$$R(W^g) = \sum_{w \in W^g} || \max(0, -w) ||_2^2 \tag{16}$$

where $W^g$ denotes the set of weight matrices in the positive dense layers of the ICNN parameterizing $g$. We train with the following hyperparameters:

- learning rate: 0.001
- optimizer: *Adam(*$\beta_1 = 0.5, \beta_2 = 0.9$*)*
- hidden layers: $[64, 64, 64, 64]$
- inner loop iterations: 10
- outer loop iterations: 25000
- batch size: 1024
- activation function: *Leaky ReLU(*$\beta = 0.01$*)*
- gradient clipping to norm: 1.0

Additionally, we employ a grid search over different levels of unbalancedness $\tau$. Results are reported in Figure 12. Introducing unbalancedness gradually improves performance up to a certain level. The results reported in Section 5.1 use $\tau = 0.85$.

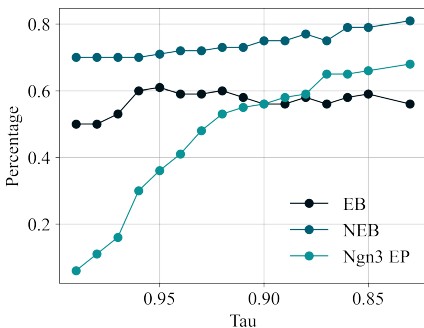

Figure 12: Effect of unbalancedness parameter $\tau$ on aggregated transition probabilities. The trend shows that introducing unbalancedness to a certain extent by lowering $\tau$ improves performance for all lineages. Metrics are computed the same way as in Table 1.

### F.5.3 PRETRAINING

We pretrain the ICNN parameterizing $f$ on the identity map as suggested in Korotin et al. (2020); Amos et al. (2023) such that $\nabla f(x) = x$ and then copy the weights to $g$ for them to be mutually inverse $\nabla f(\nabla g(x)) \approx x$ and $\nabla g(\nabla f(x)) \approx x$. Therefore, we train on $\mu \sim \mathcal{N}(0, 3)$ for $15,000$ iterations.

Additionally, for the balanced settings, we perform best model selection based upon the lowest forward Sinkhorn divergence which is a debiased estimate of the Wasserstein distance. We also empirically observe better results using this stopping criterion for the unbalanced setting.

### F.5.4 CELL TYPE TRANSITION METRIC

We follow Bastidas-Ponce et al. (2019) to obtain the ground truth of cell type transitions. We only consider cell type transitions where the target cell state is a terminal cell state $t \in T = \{$Acinar, Ductal, Alpha, Beta, Delta, Epsilon$\}$ or a union thereof. Let **ED** be the set of endocrine cell types (Alpha, Beta, Delta, Epsilon). We assume the following cell type transitions are exclusively correct (denoted by $\rightarrow$), i.e. there is no descending cell type (or set of cell types) other than the given one. We partition all considered cell type transitions into three categories.

The first set of considered transitions are endocrine branch (**ED**) transitions:

- Fev+ Alpha (**FA**) $\rightarrow$ Alpha (**A**)
- Fev+ Beta (**FB**) $\rightarrow$ Beta (**B**)
- Fev+ Delta (**FD**) $\rightarrow$ Delta (**D**)
- Fev+ Epsilon (**FE**) $\rightarrow$ Epsilon (**E**)
- **A** $\rightarrow$ **A**

- **B → B**
- **D → D**
- **E → E**

The second set of transitions are **Ngn3 EP** transitions:

- Ngn3 high early (**NE**) → **ED**
- Ngn3 high late (**NL**) → **ED**

The third set of transitions is the non-endocrine branch (**NEB**)

- Ductal (**DU**) → **DU**
- Tip (**T**) → Acinar (**AC**)
- **AC → AC**

In Table 10 we report the transition probabilities for all above-mentioned cell type transitions for OT-ICNN, UOT-ICNN, and established trajectory inference methods.

### F.5.5 EVALUATION OF COMPETING METHODS

For evaluating cell type transitions we use CellRank kernels. Here, kernels quantify transition probabilities based on vector fields. For all methods yielding velocity vectors (OT-ICNN, UOT-ICNN, scVelo, TrajectoryNet), we use the VelocityKernel. For evaluating Waddington OT (WOT), we use the WOTKernel.

**scVelo.** To infer RNA velocity with scVelo, we first selected genes measured in at least 20 cells in both unspliced and spliced transcripts. Next, cells are normalized by the median cell size, and the 2000 highly variable genes are selected. For these preprocessing steps, we used scVelo's `filter_and_normalize` function. Following, moments were calculated by the `scvelo.pp.moments` function with the settings `n_pcs=50` principal components, and `n_neighbors=30` nearest neighbors. RNA velocity was inferred using the `recover_dynamics` function implemented in scVelo.

**TrajectoryNet.** We run trajectory net with default parameters as suggested by the author's Jupyter notebooks. Specifically, `embedding=PCA`, `max_dim=10`, `max_iterations=10,000` and `vecint=1e-4`. We computed velocities by subtracting the inferred coordinates from the original coordinates in the embedding space. Since we were able to retrieve the inferred coordinates only for one time point, we set the velocities of the other time point to **0**.

**Waddington OT.** The Waddington OT results were calculated with CellRank's `WOTKernel`. For the corresponding transition matrix, we considered both inter and intra timepoint transitions. The intra timepoint transitions were quantified for each time point independently by the cell-cell nearest neighbor graph, and assigned a weight of $0.2$. Summarizing, `WOTKernel`'s `compute_transition_matrix` method was run with `growth_iters=3`, `growth_rate_key="growth_rate_init"`, `self_transitions="all"`, and `conn_weight=0.2`.

### F.5.6 VELOCITY STREAM EMBEDDING

Velocity vectors were projected onto the two-dimensional UMAP embedding using scVelo's `velocity_embedding_stream` function. To project the high-dimensional vectors, we consider the empirical displacement given by the difference of a cellular representation in the low-dimensional embedding. The displacement vector in UMAP coordinates is then defined as the expected empirical displacement under a transition matrix and corrected by the expected shift under a uniform distribution. To define the entry $(j, k)$ of the transition matrix, consider the reference cell $j$ and a neighbor $k$. The probability that cell $j$ transitions into cell $k$ is defined as the normalized Pearson correlation between the empirical displacement in the high dimensional f of the two cells, and the velocity vector of the reference cell.

## F.6 SIMULATED DATA

The simulated data consists of the union of draws of uniform distributions on $\mu_1 \sim \mathcal{U}([-0.5, 0.5] \times [-1.5, -0.5])$ and $\mu_2 \sim \mathcal{U}([4.5, 5.5] \times [-1.5, -0.5])$. Similarly, $\nu_1 \ sim\mathcal{U}([-0.5, 0.5] \times [0.5, 1.5]$ and $\nu_2 = \mathcal{U}([4.5, 5.5] \times [0.5, 1.5])$. The source distribution $\mu$ is obtained by drawing 180 samples from $\mu_1$ and 120 samples from $\mu_2$. Similarly, the target distribution $\nu$ is obtained by 180 samples from $\nu_2$ and 120 samples from $\nu_1$. To compute couplings, we use $\epsilon = 0.1$ in this case.

