# OpenReview forum: "Unbalancedness in Neural Monge Maps Improves Unpaired Domain Translation"
_ICLR.cc/2024/Conference — ICLR 2024 poster_

### Official Review · Reviewer_1iBx · 2023-10-25

**Soundness:** 2 fair
**Presentation:** 3 good
**Contribution:** 2 fair
**Rating:** 6
**Confidence:** 3

**Summary:**

This paper proposes a general framework for incorporating unbalancedness into any Monge map estimator. This paper shows that the unbalanced Monge map corresponds to the Monge map (from OT) between the reweighted source and target distributions. The proposed framework is to introduce existing Monge map estimator into these reweighted distributions. These reweighted distributions are estimated through the minibatch UOT estimator.

**Strengths:**

-	This work is overall well-written.
-	This paper proposes a general framework for extending existing OT Monge maps estimators into unbalanced cases.
-	This paper shows that the introduction of unbalancedness improves the performance of existing OT Monge map estimators in unpaired domain translation tasks.

**Weaknesses:**

-	**W1.** The main Prop 3.1-1, which provides the justification for the proposed general framework, was proved in [1] (The proof is different).
-	**W2.** The performance is only compared with the OT-counterpart and not with other UOT Monge maps estimators.

**Questions:**

-	**Q1.** What is the definition of $\tau$ in Prop 3.1? Is it the same $\tau$ in Appendix B?
-	**Q2.** What is the reason for better performance of UOT compared to OT? This paper suggest that this improvement is primarily attributed to the discrepancy in the number of samples for each corresponding cluster, e.g. $8 \leftrightarrow B$ in Fig 1. If the number of samples were similar, would the performance of UOT be comparable to OT?
-	**Q3.** The proposed framework is to optimize the OT Monge map on the estimated reweighted distribution. How does the performance of the proposed framework change according to the minibatch UOT coupling estimates, such as minibatch size and $\epsilon$ of entropic regularization, in terms of quantitative results?
- **Typo**  $\phi^{\*} (x) d \mu (x) \rightarrow \phi^{\*}(y) d \nu (y)$ in Eq (2)

**Reference**

[1] Choi, Jaemoo, Jaewoong Choi, and Myungjoo Kang. "Generative Modeling through the Semi-dual Formulation of Unbalanced Optimal Transport." arXiv preprint arXiv:2305.14777 (2023).

---

> ### Author Response · Authors · 2023-11-16
> **Response to Review 1/2**
>
> > ***The main Prop 3.1-1, which provides the justification for the proposed general framework, was proved in [1] (The proof is different).***
>
> ➤ Thanks for pointing out the similarities to the concurrent work [Choi+'2023]. However, while we agree that these results appear to be related we respectfully disagree that Point 1 of our Prop 3.1 was proved in [Choi+'2023]. To highlight that point, we analyze their theory. Since [Choi+'2023, Theorem 3.4] presents a bound on the dual sub-optimality gap, hence dealing with aspects not covered in our paper, it seems that the reviewer mentions [Choi+'2023, Theorem 3.3]. For this reason, we focus on the latter Theorem. In the following, we want to lay out the differences between our and their results.
>
> - [**Choi+'2023]'s theory is developed under stronger assumptions** than ours, which are given in the first paragraph of [Choi+'2023, Appendix A].  On the one hand, they assume the cost to be the squared Euclidean distance $c(\mathbf{x}, \mathbf{y}) = \tfrac{\tau}{2}\|\mathbf{x} - \mathbf{y}\|_2^2$ $(\tau > 0)$, while our results hold for any costs $c(\mathbf{x}, \mathbf{y}) = h(\mathbf{x} - \mathbf{y})$ with $h$ strictly convex. We hence recover the squared Euclidean distance with $h = \tfrac{\tau}{2} \|\cdot\|_2^2$. On the other hand, they assume that $\mu, \nu \in \mathcal{M}_1^+(\Omega)$ are probability measures of same total mass 1, while our results hold for any positive measures $\mu, \nu \in \mathcal{M}^+(\Omega)$ of arbitrary total mass, that might be different.
>
> - **We then clarify the correspondence between our and theirs notations.** They denote $(v^{\star, c}, v^{\star})$ the optimal potentials solving the Unbalanced Kantorovich Dual Problem (UDKP), whereas we denote them by $(f^\star, g^\star)$. Furthermore, they denote $\Psi$ the entropy function associated to the divergence used to measure the deviation from the marginals, whereas we denote it $\phi$. Finally, in the same way as us, they note $\tilde{\mu} = \bar{\phi}(-f^\star)\cdot\mu$ and $\tilde{\nu} = \bar{\phi}(-g^\star)\cdot\mu$.
>
> - **We now highlight that point 1 of Prop 3.1 is not shown in [Choi+'2023].** Indeed, [Choi+'2023, Theorem 3.3] proves that if we let $T^\star$ be the Monge map between $\tilde{\mu}$ and $\tilde{\nu}$, which exists because of the assumptions on the costs and the fact that $\tilde{\mu} \ll \mu \ll \mathcal{L}_d $, then one has:
>
> $$
> \forall \mathbf{x} \in \mathcal{X}, \textrm{  } T(\mathbf{x}) \in \arg \inf_{\mathbf{y} \in \mathcal{Y}} c(\mathbf{x}, \mathbf{y}) - g^\star(\mathbf{y})
> $$
>
> However, they do not show that $\pi_\mathrm{UOT}$, the unbalanced UOT plan  between $\mu$ and $\nu$, is supported on the graph of that map, i.e.  $\pi_\mathrm{UOT} = (\mathrm{Id}, T^\star)\sharp\tilde{\mu}$ . Therefore, their statement do not show that $p_1 \sharp \pi_\mathrm{UOT} = \tilde{\mu}$ and $p_2 \sharp \pi_\mathrm{UOT} = \tilde{\nu}$, i.e. that $\pi_\mathrm{UOT}$  has marginals $\tilde{\mu}$ and $\tilde{\nu}$, which is the result stated in Point 1 of our Prop. 3.1. Moreover, in their proof, we don't understand how they obtain that $\tilde{\mu}$ and $\tilde{\nu}$ have the same total mass, which must be justified before considering the Monge map $T^\star$ between $\tilde{\mu}$ and $\tilde{\nu}$. In our proof, we obtain it by showing that $\tilde{\mu}$ and $\tilde{\nu}$ are the marginals of $\pi_\mathrm{UOT}$, so the equality of masses follows from Fubini's Theorem.
>
> - **Proposition 3.1 (Point 1 and 2) is in fact a more general version of their [Choi+'2023, Theorem 3.3].** Indeed, for all costs $c(\mathbf{x}, \mathbf{y}) = h(\mathbf{x} - \mathbf{y})$, we show that
>
> $$
> \forall \mathbf{x} \in \mathcal{X}, \textrm{  } T^\star(\mathbf{x}) = \mathrm{Id} - \nabla h^\star \circ \nabla f^\star(\mathbf{x})
> $$
>
> and that $\pi_\mathrm{UOT}$ is unique, and characterized by $\pi_\mathrm{UOT} = (\mathrm{Id}, T^\star)\sharp\tilde{\mu}$. This result then allow us to extend their result for $c(\mathbf{x}, \mathbf{y}) = \tfrac{\tau}{2} \|\mathbf{x} - \mathbf{y}\|_2^2$. As we mentioned in Remark. 3.2, in this case:
>
> $$
> T^\star(x) = \mathbf{x} - \nabla f^\star(\mathbf{x}) = \nabla \varphi^\star(\mathbf{x})
> $$
>
> where $\varphi^\star : \mathbf{x} \mapsto \|\mathbf{x}\|_2^2 - f^\star(\mathbf{x})$ is convex and unique. We then obtain an unbalanced generalization of [Brenier'1987]'s Theorem, stating that as in the balanced OT case, when the source measure is a density, the unbalanced OT plan is unique supported on the graph of a convex function.
>
> We would be happy to answer any questions that might arise and clarify if some arguments were not clear.

---

> > ### Author Response · Authors · 2023-11-16
> > **Response to Review 2/2**
> >
> > > ***The performance is only compared with the OT-counterpart and not with other UOT Monge maps estimators.***
> >
> > ➤ While we would like to highlight that the main message of this paper is that unbalanced Monge maps are favorable over balanced Monge maps, we hope that our newly added benchmark for the sciPlex dataset addresses the reviewer's concern.
> >
> > > ***Q1. What is the definition of $\tau$ in Prop 3.1? Is it the same in Appendix B?***
> >
> > ➤ Yes, thanks for pointing this out. We moved the definition to the main text in Section 2.
> >
> > > ***Q2. What is the reason for better performance of UOT compared to OT? This paper suggest that this improvement is primarily attributed to the discrepancy in the number of samples for each corresponding cluster, e.g.
> >  in Fig 1. If the number of samples were similar, would the performance of UOT be comparable to OT?***
> >
> > ➤ Thanks for this question. We've decided to add a paragraph in Section 3.1 to further elaborate on the merits of UOT. Specifically, through batch-wise sub-sampling discrepancies are very likely to appear at a batch level. This means that UOT can help stabilize training even when there is no distribution shift between source and target.
> >
> > > ***Q3. The proposed framework is to optimize the OT Monge map on the estimated reweighted distribution. How does the performance of the proposed framework change according to the minibatch UOT coupling estimates, such as minibatch size and
> >  of entropic regularization, in terms of quantitative results?***
> >
> > ➤ As mentioned in the general comment, we have added additional results where we empirically benchmark the effects of batch size and entropic regularization on EMNIST (Appendix C.4).
> >
> > ***
> >
> > [Choi+'2023] Choi, Jaemoo, Jaewoong Choi, and Myungjoo Kang. "Generative Modeling through the Semi-dual Formulation of Unbalanced Optimal Transport." arXiv preprint arXiv:2305.14777 (2023).
> >
> > [Brenier’1987] Yann Brenier. Décomposition polaire et réarrangement monotone des champs de vecteurs. CR Acad.
> > Sci. Paris Sér. I Math., 305, 1987.

---

> > > ### Comment · Reviewer_1iBx · 2023-11-19
> > >
> > > Thank you for the response. The clarification of theoretical contribution and the additional experimental results (Appendix C.4 and Table 5) have been helpful in addressing my concerns, (although CIFAR-10 generation results are not that impressive). Hence, I raised my rating to 6.

---

### Official Review · Reviewer_Rczx · 2023-10-26

**Soundness:** 3 good
**Presentation:** 3 good
**Contribution:** 3 good
**Rating:** 6
**Confidence:** 3

**Summary:**

This paper studies the methodological aspect of optimal transport (OT) under the practical application scenarios for domain translation, e.g., the mass proportion is imbalanced for different domains. The main pursuit is to approximately estimate the dual OT distance via the neural networks with significant capacity and provide an explicit retrieval form for the OT plan/map. To overcome the limitation in marginal distribution conservation, this paper introduces unbalanced OT to obtain a relaxed plan and then recovers the balanced OT from a reweighting view. Theoretical results are provided to connect the re-weighted OT and UOT and ensure the feasibility of recovering the Monge map from neural UOT. Experiments are conducted from several perspectives, which validates the effectiveness of the proposed method.

**Strengths:**

+ This paper is well-written and easy to follow; necessary justifications and discussions are provided around the technical parts.
+ The proposed method is well-motivated and reasonable; the main difficulty stated in introduction is properly addressed with the proposed method and theoretical analysis.
+ Diverse experiments are conducted to validate the proposed method from different perspectives and tasks, where the results are generally convincible.

**Weaknesses:**

- Some related concepts and fields are omitted, which should also be discussed appropriately.
- The limitations of the proposed method should be discussed.
- The experiments can be enhanced by considering more practical and challenging problems.

**Questions:**

1. As far as I understand this paper, the basic goal is to address the potential mismatch induced by the imbalanced weights of different domains; more precisely, the imbalance means the mass proportion of the ideal transport pairs is different. Such a scenario is also analogous to the label shift problem, which is usually considered in OT methodology. To address this problem, there are two natural and common solutions: relaxation (i.e., UOT) or reweighting (i.e., adjusting the marginal distribution) [r1, r2]. Since this work adopts relaxation as a solution (while also noting that the idea of reweighting is also implicitly shown in Sec. 3.1), I think more justification and discussion on the related fields are highly expected.

2. Based on the reweighting solution mentioned above, can the imbalanced mass problem be addressed by detecting the degree of shifting mass (i.e., estimating the weights in Sec. 3.1) and solving the reweighting (balanced) OT? If it is feasible, what will be the advantages and weaknesses of the proposed methods?

3. As the basic problem is similar to label shift, the experiments on related problems are highly appreciated. For example, the generalized label shift scenario [r1] and partial domain adaptation [r2] (which can be taken as an extreme scenario for label shift), where the comparison methods for corresponding tasks should also be carefully considered.

4. Minor points:

4.1) Page 3, typo in the reweighting formulation of $\tilde{\nu}$ in Sec. 3.1.

4.2) Point 2 in Prop. 3.1, the definition of $\mathcal{L}_d$ is not provided.

[r1] Rakotomamonjy, Alain, et al. "Optimal transport for conditional domain matching and label shift." Machine Learning (2022): 1-20.

[r2] Gu, Xiang, et al. "Adversarial reweighting for partial domain adaptation." Advances in Neural Information Processing Systems 34 (2021): 14860-14872.

---

> ### Author Response · Authors · 2023-11-16
> **Response to Review 1/2**
>
> > ***Some related concepts and fields are omitted, which should also be discussed appropriately.***
>
> ➤ As far as we understand, this comment is mainly about the field of domain **adaptation**. We first give definitions domain of  **adaptation** and **translation**, which the remaining response builds upon, and would like to ask the reviewer for confirmation:
> - Domain **adaptation** is concerned with learning domain-invariant representations or transformations to train robust **classification** models that can generalize across domains.
> - Domain **translation**, on the other hand, concentrates on training a **generative** map to translate the source domain data into the style and distribution of the target domain, e.g. inspired by reducing the total cost incurred by this translation. The goal of domain translation is to learn the mapping between domains rather than performing a prediction task.
>
> How to leverage UOT in the domain **adaptation** task is thoroughly discussed in [Fartras+'2021], which we also refer to as an application of UOT in Section 1 ("The practical significance of unbalancedness in discrete OT has been demonstrated in various applications, e.g. in video registration (Lee et al., 2020), computer vision (Plaen et al., 2023), or domain adaptation (Fatras et al., 2021)"). However, our work is concerned with learning Monge maps for domain **translation**, and hence our framework is not directly applicable to domain **adaptation**. For domain **translation**, we believe we cover all important concepts and related fields.
>
> > ***The limitations of the proposed method should be discussed.***
>
> ➤ As mentioned in the general comment, we now discuss these in Appendix B.3 .
>
> > ***The experiments can be enhanced by considering more practical and challenging problems.***
>
> ➤ As mentioned in the general comment, we have added another real-world application in a different perturbation task on the single-cell data measured in [Srivatsan 2020] as well as a benchmark on CIFAR-10 image generation. What more challenging datasets would you be interested in seeing?
>
> > ***As far as I understand this paper, the basic goal is to address the potential mismatch induced by the imbalanced weights of different domains; more precisely, the imbalance means the mass proportion of the ideal transport pairs is different. Such a scenario is also analogous to the label shift problem, which is usually considered in OT methodology. To address this problem, there are two natural and common solutions: relaxation (i.e., UOT) or reweighting (i.e., adjusting the marginal distribution) [r1, r2]. Since this work adopts relaxation as a solution (while also noting that the idea of reweighting is also implicitly shown in Sec. 3.1), I think more justification and discussion on the related fields are highly expected.***
>
> ➤ In the following, we reply to the question in three parts. For the first part of the question regarding imbalanced weights of different domains (in the following referred to as label imbalance), we assume we have access to labeled data. We agree that unbalancedness helps to mitigate class imbalances, as demonstrated in multiple examples, e.g. in the EMNIST experiments. Yet, we would like to highlight that unbalancedness acts on batches, and hence there does not have to be a label imbalance on the full dataset for unbalancedness to have a positive effect, as within one batch, the label distribution might not reflect the label distribution of the full dataset. Moreover, we would like to highlight that unbalancedness is not only able to adapt label imbalances but also automatically discard outliers.
>
> For the second part of the question, we fully agree that rescaling the marginal distribution is an option to address class imbalance. Yet, this assumes access to labels. In many real-world scenarios, data is not labeled, or the quality of the labels is insufficient. An example of the latter is cell-type labeling in single-cell genomics, which clusters cells into disjoint classes. In fact, cells are known to evolve continuously between clusters. With respect to image labels, the labels considered in the CelebA experiments (e.g. glasses vs. no glasses) might be clear. In contrast, labels for hair color might be less obvious, and hence reweighting the distribution accordingly is difficult.

---

> ### Author Response · Authors · 2023-11-16
> **Response to Review 2/2**
>
> > ***Based on the reweighting solution mentioned above, can the imbalanced mass problem be addressed by detecting the degree of shifting mass (i.e., estimating the weights in Sec. 3.1) and solving the reweighting (balanced) OT? If it is feasible, what will be the advantages and weaknesses of the proposed methods?***
>
> ➤ As discussed in more detail above, there are three limitations to this approach: First, there might not be sufficient information to rescale the marginals (e.g. due to poor or non-existent labels). Second, unbalancedness also compensates for imbalances within one batch, which is also likely to happen in class balance on the full dataset. Third, outliers cannot be automatically discarded in this way. An advantage would be the independence of the unbalancedness parameters, which, as discussed, is not necessarily straightforward.
>
> > ***As the basic problem is similar to label shift, the experiments on related problems are highly appreciated. For example, the generalized label shift scenario [r1] and partial domain adaptation [r2] (which can be taken as an extreme scenario for label shift), where the comparison methods for corresponding tasks should also be carefully considered.***
>
> ➤ As mentioned above, (unbalanced) neural Monge maps in general cannot be applied to the domain adaptation task straight-forwardly.
>
> > ***Point 2 in Prop. 3.1, the definition of $\mathcal{L}_d$ is not provided.***
>
> ➤ We define $\mu \ll \mathcal{L}_d$ at the start of Section 2 as $\mu$ absolutely continuous w.r.t. the Lebesgue measure.
>
> ***
> [Srivatsan+'2020] Srivatsan, Sanjay R., et al. "Massively multiplex chemical transcriptomics at single-cell resolution." Science 367.6473 (2020): 45-51.
>
> [Fartras+'2021] Kilian Fatras, Thibault S\'ejourn\'e, Nicolas Courty, and R\'emi Flamary. "Unbalanced minibatch optimal
> transport; applications to domain adaptation" (2021).

---

> > ### Comment · Reviewer_Rczx · 2023-11-21
> > **Thank you for your responses.**
> >
> > I appreciate the authors for their detailed responses to the concerns. After checking the rebuttal and revised manuscript, the major concerns are addressed. Overall, this paper is well-written and easy to follow, where motivation and insights are also clearly presented. Therefore, I keep my positive score for acceptance.

---

### Official Review · Reviewer_qjsW · 2023-10-31

**Soundness:** 3 good
**Presentation:** 3 good
**Contribution:** 3 good
**Rating:** 6
**Confidence:** 3

**Summary:**

The paper explores the role of introducing unbalancedness into neural Monge maps within the framework of optimal transport (OT). The authors investigate whether this unbalancedness can enhance the performance of unpaired domain translation tasks. Through a mix of theoretical analysis and empirical validation, the paper shows that incorporating unbalancedness leads to significant improvements in terms of cost-efficiency and generalization across domains.

Contributions:

A modified OT framework that accommodates unbalancedness.
New neural estimators designed for unbalanced Monge maps.
Empirical evidence demonstrating the advantages of the modified approach in real-world applications like single-cell biology and computer vision.
The paper thus offers a novel approach to improving unpaired domain translation by tweaking the conventional OT framework and validating its effectiveness

**Strengths:**

- Novelty: The paper introduces the novel concept of "unbalancedness" into neural Monge maps, which is a fresh angle in the well-studied field of optimal transport.
- Theoretical and Empirical Validation: The work combines both theoretical reasoning and empirical results, strengthening the validity of its claims.
- Practical Impact: The paper demonstrates the utility of its approach in real-world applications, such as single-cell biology and computer vision, indicating its relevance beyond theoretical considerations.
- Methodological Rigor: The research methodology appears to be sound, involving both the development of a new framework and neural estimators, as well as their validation on synthetic and real-world data.
- Broad Applicability: The issue of unpaired domain translation is relevant in multiple fields, and the paper's contributions could be generalized to other domains, increasing its impact.

**Weaknesses:**

- Lack of External Benchmarks: Without a comparison to existing methods or frameworks, it's difficult to assess how much of an improvement the proposed approach offers.
- Complexity: Introducing unbalancedness into neural Monge maps could add computational or conceptual complexity, which may not be fully addressed in the paper.
- Dependency on Empirical Validation: While the paper does include empirical validation, its contributions could be further strengthened with more diverse datasets or a broader set of real-world applications.

**Questions:**

What are the limitations of introducing unbalancedness into Monge maps, and are there scenarios where this approach may not be beneficial?

How does the proposed unbalanced neural Monge map approach compare with existing methods in terms of efficiency and accuracy?

---

> ### Author Response · Authors · 2023-11-16
> **Response to Review**
>
> > ***Lack of External Benchmarks: Without a comparison to existing methods or frameworks, it's difficult to assess how much of an improvement the proposed approach offers.***
>
> ➤ We compare to competing methods for single-cell trajectory inference (Appendix C.5) and image translation (Table 3, Appendix C.1). For the perturbation modeling task on 4i data, the Monge Gap is the current state-of-the-art to the best of our knowledge. Hence, we continue this line of research by comparing the balanced Monge Gap to our proposed unbalanced version as stated in Section 5.2. ("Since Uscidda & Cuturi (2023) show that OT-MG outperforms OT-ICNN in this task, we continue this line of research by applying unbalanced OT-MG (UOT-MG) to predict cellular responses to 35 drugs from data profiled with 4i technology (Gut et al., 2018).")
>
> We agree with the reviewer that competing unbalancedness estimators were not considered. We hope to overcome this limitation with the newly added benchmark for the sciPlex dataset.
>
> > ***Complexity: Introducing unbalancedness into neural Monge maps could add computational or conceptual complexity, which may not be fully addressed in the paper.***
>
> ➤ We agree with the reviewer and as mentioned in the general comment have now added a discussion of the computational complexity introduced by our framework in Appendix B.3. We discuss precisely the cost of incorporating unbalancedness in each Monge map estimator.
>
> > ***Dependency on Empirical Validation: While the paper does include empirical validation, its contributions could be further strengthened with more diverse datasets or a broader set of real-world applications.***
>
> ➤ As mentioned in the general comment, we have added another real-world application in a different perturbation task on single-cell data measured in [Srivatsan+'2020] as well as a benchmark on CIFAR-10 image generation. What other datasets or tasks would you be interested in seeing?
>
> > ***What are the limitations of introducing unbalancedness into Monge maps, and are there scenarios where this approach may not be beneficial?***
>
> ➤ Thanks for this question. As mentioned in the general comment, we've added a section on the limitations of our framework.
>
> > ***How does the proposed unbalanced neural Monge map approach compare with existing methods in terms of efficiency and accuracy?***
>
> ➤ In the following, we assume that the question addresses the comparison of efficiency and accuracy to other unbalancedness estimators. The comparison of UOT-ICNN with [Lübeck+'2022] is discussed in length in section 4. In particular, we state "Third, our approach is computationally more efficient [than the approach by Luebeck et al.], as our method requires the
> computation of at most one discrete unbalanced OT plan, while the algorithm proposed in [Lübeck+'2022] includes the computation of two discrete unbalanced OT plans." A comparison of UOT-ICNN with other approaches that model unbalancedness (e.g. [Yang+'2019] and [Choi+'2023]) with respect to efficiency is not possible due to the different modeling assumptions. Similarly, comparisons with respect to the efficiency (complexity) of UOT-MG and UOT-FM are not possible. Regarding accuracy, as mentioned in the general comment we've now added a new benchmark comparing UOT-ICNN to related unbalancedness estimators.
>
> ***
> [Yang+'2019] Karren D. Yang and Caroline Uhler. "Scalable unbalanced optimal transport using generative adversarial networks", (2019).
>
> [Srivatsan+'2020] Srivatsan, Sanjay R., et al. "Massively multiplex chemical transcriptomics at single-cell resolution." Science 367.6473 (2020): 45-51.
>
> [Lübeck+'2022] Frederike L\"ubeck, Charlotte Bunne, Gabriele Gut, Jacobo Sarabia del Castillo, Lucas Pelkmans, and
> David Alvarez-Melis. "Neural unbalanced optimal transport via cycle-consistent semi-couplings", arXiv preprint arXiv:2209.15621 (2022).

---

### Official Review · Reviewer_LDyN · 2023-11-03

**Soundness:** 3 good
**Presentation:** 3 good
**Contribution:** 3 good
**Rating:** 6
**Confidence:** 4

**Summary:**

The paper introduces unbalancedness into any balanced Monge map estimator based on optimal transport (OT). Moreover, it demonstrates, both theoretically and experimentally, that this can enhance the performance of unpaired domain translation tasks.

**Strengths:**

1. This paper is well writing and is well organized, and proposed method is well-motivated and effective.
2. This work innovatively integrates unbalanced optimal transport into the existing Monge map estimator, achieving competitive results in various domain translation generative tasks.
3. Sufficient theoretical and experimental evidence is provided in the main paper and appendix.

**Weaknesses:**

1. In the single-cell trajectory inference and image translation experiments in Sections 5.1 and 5.3 (Table 1, 2, and 4), the article compared the performance of OT and unbalanced OT. Were there comparisons with other state-of-the-art methods?
2. The experiments in Table 3 on the CelebA dataset show that the proposed method performs suboptimally FID score in comparison to UVCGAN [1], which was introduced in 2022. Further explanation is required to address this disparity.
3. OT-FM [2] has conducted numerous experiments in image generation tasks and outperforms many GAN-based and diffusion-based methods. Have you considered experimenting with UOT-FM on the same tasks for further validation of the method's effectiveness?

[1] Lipman Y, Chen R T Q, Ben-Hamu H, et al. Flow matching for generative modeling.
[2] Torbunov D, Huang Y, Yu H, et al. Uvcgan: Unet vision transformer cycle-consistent gan for unpaired image-to-image translation.

**Questions:**

Please see the above weakness part.

---

> ### Author Response · Authors · 2023-11-16
> **Response to Review**
>
> > ***In the single-cell trajectory inference and image translation experiments in Sections 5.1 and 5.3 (Table 1, 2, and 4), the article compared the performance of OT and unbalanced OT. Were there comparisons with other state-of-the-art methods?***
>
> ➤ Yes, for single-cell trajectory we compare to competing methods in Appendix C.5 as mentioned at the end of Section 5.1 . For EMNIST, we chose not to compare to any competing methods, as this is meant as an illustrative example to show the difference between FM, OT-FM, and UOT-FM. For CelebA, we compare to competing methods in Table 3 (latent space) and Appendix C.1 (pixel-space). For Table 4, we chose to evaluate the learned transport cost only for the FM-based methods as the difference between them is the main result we want to emphasize.
>
> > ***The experiments in Table 3 on the CelebA dataset show that the proposed method performs suboptimally FID score in comparison to UVCGAN [1], which was introduced in 2022. Further explanation is required to address this disparity.***
>
> ➤ We argue that UOT-FM should be viewed as a **principled method** for unpaired image translation and compared to other such methods like CycleGAN [Zhu+'2017] or [Korotin+'2022], which UOT-FM significantly outperforms in all tasks as detailed in Table 3 and Appendix C.1 .
>
> As mentioned in Appendix E.3 UVCGAN [Torbunov+'2022] is a **highly optimized** version of CycleGAN that uses an *improved architecture* (UNet + Vision Transformer at the bottleneck layer) and **self-supervised pre-training**. Due to computational reasons, we pre-compute all latent embeddings with the Stable Diffusion encoder and thus are also not benefitting from random cropping during training, which UVCGAN leverages as stated in Appendix F.2.1 . We believe a similar kind of optimization could also be done for UOT-FM to significantly improve its performance.
>
> > ***OT-FM [2] has conducted numerous experiments in image generation tasks and outperforms many GAN-based and diffusion-based methods. Have you considered experimenting with UOT-FM on the same tasks for further validation of the method's effectiveness?***
>
> ➤ Originally, we chose not to benchmark on generative modeling tasks as the main focus of the paper is **domain translation**. However as mentioned in the general comment, we have now added an experiment on CIFAR-10 image generation.
>
> ***
>
> [Zhu+'2017] Jun-Yan Zhu, Taesung Park, Phillip Isola, and Alexei A Efros. "Unpaired image-to-image translation
> using cycle-consistent adversarial networks". In IEEE International Conference on Computer Vision 2017 (ICCV), 2017.
>
> [Korotin+'2022] Alexander Korotin, Daniil Selikhanovych, and Evgeny Burnaev. "Neural optimal transport". 2022. URL https://arxiv.org/abs/2201.12220.
>
> [Torbunov+'2022] Dmitrii Torbunov, Yi Huang, Haiwang Yu, Jin Huang, Shinjae Yoo, Meifeng Lin, Brett Viren, and Yihui Ren. "Uvcgan: Unet vision transformer cycle-consistent gan for unpaired image-to-image translation", 2022.

---

### Author Response · Authors · 2023-11-16
**General Comment**

We would like to thank all reviewers for their time and thoughtful efforts in providing detailed and constructive feedback. We appreciate their recognition of the clarity in writing (*LDyN, Rczx, 1iBx*), and overall assessment of soundness, presentation, and contribution (*LDyN, qjsW, Rczx*). Specifically, we thank the reviewers for acknowledging the strong theoretical foundation and extensive experimental evidence (*LDyN, Rczx, 1iBx*), as well as their appreciation of a sound motivation (*LDyN, Rczx, 1iBx*) and generality of our proposed unbalanced neural Monge map framework (*qjsW, Rczx, 1iBx*).

We want to highlight the changes we made to our submission and some of the joint questions here while additionally, we address each raised issue with individual replies for each reviewer.


**New experiments.** In response to the reviews, we've added three more experiments:
- Generative modeling on CIFAR-10 comparing UOT-FM to OT-FM. We show that UOT-FM offers faster convergence and better performance compared to OT-FM. We report FID, and transport cost at the last iteration (Section 5.3) as well as their convergence over training in Appendix C.7 together with random samples of generated images (Appendix C.8). UOT-FM achieves state-of-the-art performance on CIFAR-10 for simulation-free neural ODE training algorithms.
- We benchmark UOT-ICNN against competing neural unbalancedness estimators, [Lübeck+'2022] and [Yang+'2019], on the sciPlex perturbation dataset [Srivatsan+'2020]. We decided to choose a new dataset for this benchmark to increase the variety of considered datasets requested by some reviewers. (Appendix C.6)
- We empirically benchmark the effect of the batch size and $\varepsilon$ for results on the EMNIST dataset (Appendix C.4). Our results align with theoretical results (as $\varepsilon$ is increased UOT-FM approaches FM) and previous studies (a fairly low batch size already yields competitive results). (Appendix C.4)

**Limitations.** We agree with the reviewers, that we did not sufficiently discuss the limitations of our proposed method. We've now added this in Appendix B.2, which we reference in our discussion (Section 6). We consider the choice of the additional hyperparameter $\tau = (\tau_1, \tau_2)$ as the main limitation of our proposed framework. In line with this, we've also added Appendix B.3 discussing the added computational cost of our framework, which is negligible in most cases.

**Competing neural unbalanced methods.** We would like to emphasize that the main message of this paper is to convey that unbalanced Monge maps are favorable compared to balanced Monge maps, and we propose a method to do so for any neural Monge map estimator. Hence, we choose not to benchmark UOT-ICNN against UOT-MG or UOT-FM, but highlight the difference with their respective balanced counterpart, while restricting ourselves to domain-specific competitors. Moreover, our framework is the first one applicable to *any* neural Monge map estimator and as such there is no competing method that also is applicable to the three estimators we benchmark with. However, we agree with the reviewer that competing unbalancedness estimators were not considered. We hope to overcome this limitation with the newly added benchmark of the sciPlex dataset.

***
[Lübeck+'2022] Frederike Lübeck, Charlotte Bunne, Gabriele Gut, Jacobo Sarabia del Castillo, Lucas Pelkmans, and
David Alvarez-Melis. "Neural unbalanced optimal transport via cycle-consistent semi-couplings", arXiv preprint arXiv:2209.15621 (2022).

[Yang+'2019] Karren D. Yang and Caroline Uhler. "Scalable unbalanced optimal transport using generative adversarial networks", (2019).

---

### Meta-Review · Area_Chair_szbW · 2023-12-08

**Metareview:**

The paper proposes to take into account unbalancedness into any balanced Monge map estimator based on optimal transport.
They propose a framework with algorithm and theoretical support for that.

Overall, while the paper does not have outstanding scores, reviewers found that the paper introduces
new ideas and algorithms that can be useful for the community, and I concur with those points. As such, they believe that the
paper deserves to be presented at iclr.

**Justification For Why Not Higher Score:**

The paper  does not propose a incredibly new idea (they build on the neural monge map literature to obtain unbalanced neural monge map)

**Justification For Why Not Lower Score:**

the paper introduces novel methodology that is useful for the community

---

### Decision · Program_Chairs · 2024-01-16

Accept (poster)